# MulTaBench: Benchmarking Multimodal Tabular Learning with Text and Image

Alan Arazi [* 1 2]   Eilam Shapira [* 1]   Shoham Grunblat [1]   Mor Ventura [1]   Elad Hoffer [3]   Gioia Blayer [4]
David Holzmüller [4]   Lennart Purucker [2 5]   Gaël Varoquaux [4 6]   Frank Hutter [2 5 7]   Roi Reichart [1]

## Abstract

Tabular Foundation Models have recently established the state of the art in supervised tabular learning. However, they lack native support for unstructured modalities such as text and image, and rely on frozen, pretrained embeddings to process them. We show that tuning the embeddings to the task improves performance on established Multimodal Tabular benchmarks. We introduce MulTaBench, a benchmark of 40 datasets, split equally between image-tabular and text-tabular tasks. We focus on predictive tasks where the modalities provide complementary predictive signal, and where generic embeddings lose critical information, necessitating Target-Aware Representations that are aligned with the task. We demonstrate that the gains from target-aware representation tuning generalize across both text and image modalities, several tabular learners, encoder scales, and embedding dimensions. MulTaBench constitutes the largest image-tabular benchmarking effort to date, enabling the research of novel architectures which incorporate target-aware representations, paving the way for the development of novel Multimodal Tabular Foundation Models.

## 1. Introduction

Tabular Foundation Models (TFMs) (Van Breugel & Van Der Schaar, 2024; Hollmann et al., 2022; 2025) have recently emerged as the state of the art (SOTA) for supervised tabular learning (Erickson et al., 2025). However, the best-performing TFMs (Grinsztajn et al., 2026; Qu et al., 2026) are trained exclusively on structured numerical data, making them fundamentally unimodal: unstructured inputs must be preprocessed via external embedding models, with no unified support for modalities such as text and image.

Yet, in many high-impact domains, tabular problems are multimodal: e-commerce listings, social media feeds, and medical health records combine image and text with numerical features. While early work has begun extending TFMs to integrate text (Arazi et al., 2025; Spinaci et al., 2025), these extensions often compromise the model's core tabular performance, and inherent support for visual modalities remains entirely absent. One might turn to Large Language and Vision-Language Models (LLMs/VLMs), which natively process unstructured inputs, but they are not suited for the inductive biases of tabular data; specifically, they are unoptimized for the relational structure (Fang et al., 2024) and are suboptimal for numerical features (Van Breugel & Van Der Schaar, 2024). Addressing these limitations requires architectures that combine the numerical precision of TFMs while maintaining the rich input handling of multimodal foundation models. However, evaluating such a unified approach is difficult because the diverse nature of tasks within Multimodal Tabular Learning (MMTL) is not yet fully characterized; existing benchmarks (Shi et al., 2021; Lu et al., 2023; Kim et al., 2024; Tang et al., 2024b; Mráz et al., 2025) primarily highlight the coexistence of modalities, unintentionally grouping together problems that require fundamentally different modeling solutions.

To characterize these problems, we observe that tabular models require inputs to be represented as feature columns, so high-dimensional images and texts must be compressed into compact representations. Consequently, embeddings act as lossy summaries, as they capture only a fraction of the raw input's information by design (Weller et al., 2025). In order to generalize well, pretrained embedding models are optimized for broad semantic content, such as distinguishing an X-ray from a mammogram, at the expense of fine-grained details like precise size estimations or localized anomalies (Pantazopoulos et al., 2024; Li et al., 2025). While this compression is effective for global semantic mapping, it fails to preserve the specialized signals required for fine-grained MMTL tasks. We thus advocate for the need for Target-Aware Representations (TAR): embeddings that are tuned to the target and, ideally, to the other modalities.

[1]Technion – Israel Institute of Technology [2]Prior Labs [3]NVIDIA [4]SODA Team, INRIA Saclay, Palaiseau [5]University of Freiburg [6]Probabl [7]ELLIS Institute Tübingen. Correspondence to: Alan Arazi <alanarazi7@gmail.com>, Eilam Shapira <eilam.shapira@gmail.com>.

*Proceedings of the $2^{nd}$ ICML Workshop on Foundation Models for Structured Data*, Seoul, South Korea. 2026. Copyright 2026 by the author(s).

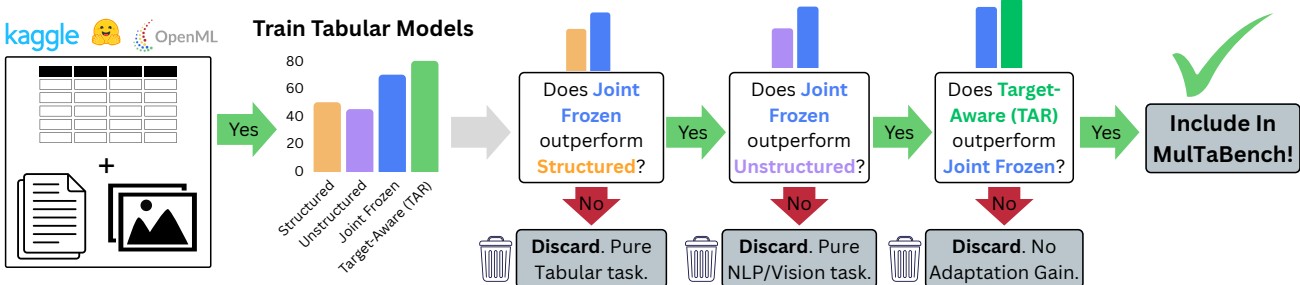

*Figure 1.* The MulTaBench Curation Pipeline. Datasets are included if they require *Joint Signal* and *Task-awareness*.

Consider, for example, the task of pneumonia detection from a patient record combining age and smoking status with chest X-ray images. We argue that to study MMTL, a dataset should satisfy two properties: (1) *Joint Signal*, where each modality provides complementary information that contributes to the overall predictive performance, and (2) *Task-awareness*, where task-agnostic representations fail to capture the details required for a given objective. In our example, both the X-ray and the clinical profile offer unique, complementary information, and steering the image embedding to detect subtle signs of inflammation in the lungs should improve diagnostic accuracy.

To translate these theoretical properties into a measurable test, we develop an algorithmic pipeline that quantifies whether a dataset complies with the aforementioned requirements. This approach approximates these properties by evaluating each task across a broad suite of tabular learners, ranging from light GBDTs to SOTA TFMs. To evaluate for *Joint Signal*, we demand a performance drop when either modality is removed, verifying that each input strengthens the predictive power. For *Task-awareness*, we finetune the encoder's last 3 layers with LoRA (Hu et al., 2021) on the prediction target as a preprocessing step, and we expect these representations to outperform frozen ones when passed to tabular models. Crucially, our experiments confirm that TAR outperform frozen embeddings across established MMTL benchmarks; however, we find that the magnitude of these gains is highly dataset-dependent, suggesting they represent distinct classes of MMTL tasks.

Building on this framework, we introduce **MulTaBench**, a benchmark of 40 datasets balanced between image-tabular and text-tabular tasks, as well as classification and regression objectives. To ensure a comprehensive evaluation, the benchmark incorporates a wide range of sample sizes and feature counts, while spanning a diverse set of domains to capture the heterogeneity of real-world multimodal tabular data. MulTaBench represents the largest image-tabular benchmarking effort to date, and the first MMTL benchmark to explicitly prioritize datasets requiring task-aware

representations. Demonstrating the robustness of our curation criteria, we show that the gains from target-aware tuning generalize consistently across a diverse suite of independent tabular learners, encoder scales, and embedding dimensions. These findings suggest that designing novel architectures which contextualize the representations of unstructured modalities can push the boundaries of MMTL, and we believe that MulTaBench would be instrumental for developing true Multimodal TFMs. See Appendix A for extended introduction, and Appendix B for Related Work.

## 2. Benchmarking MMTL

### 2.1. Desiderata for Multimodal Tabular datasets

**Joint Signal.** Following the principle in Mráz et al. (2025), we require each modality to carry independent signal about the target, so the joint predictive performance exceeds the union of unimodal performances. In the pneumonia case, the X-ray encodes spatial lung patterns, while age and smoking status convey clinical risk factors that provide information invisible in pixels. This criterion could optionally capture cross-modal interactions, where one modality might only become discriminative once conditioned on the other. For instance, increased reticular markings may signal acute infection in non-smokers, yet merely represent baseline chronic changes in a long-term smoker; the visual feature only becomes discriminative when conditioned on the tabular history. A modality can fail this criterion if it carries no signal, or if its signal is already captured by another modality and thus provides no predictive gain.

**Task-awareness** A task exhibits *Task-awareness* when the predictive signal is latent in the raw input at a level of granularity that differs from the modality's global semantic meaning. Because general-purpose encoders are optimized to preserve high-level properties while discarding low-level variance, such as exact wording (Weller et al., 2025) or fine-grained spatial textures (Pantazopoulos et al., 2024), they often discard the specific nuances required for MMTL. Recovering this signal necessitates TAR, which steer the

representation to focus on the details relevant to the specific target. In our pneumonia example, a generic model might identify the scan's global anatomy, whereas TAR would preserve the tiny visual patterns in the lung tissue that are key for diagnosis. Conversely, a task lacks *Task-awareness* if the predictive signal is coarse enough to be captured by task-agnostic embeddings; for instance, if the objective is simply to categorize the scan type rather than identify a specific pathology, TAR would provide no advantage.

## 2.2. The Curation Pipeline

To bridge the gap between the theoretical desiderata and the empirical curation, we establish an evaluation protocol based on 4 experimental conditions, as summarized in Figure 1 and Table 1 in Appendix C. The conditions vary by the features included and the specific representation of the unstructured modalities. Our approach intentionally entangles task properties with algorithmic solutions in order to isolate datasets that align with our criteria and that current models struggle with. Embeddings are extracted using *e5-v2-small* (Wang et al., 2024) for texts and *DINO-v3-small* (Siméoni et al., 2025) for images. To implement our proposed TAR condition, we finetune the last 3 layers on the prediction target using LoRA. Crucially, this adaptation is performed as a specialized preprocessing step without the structured features and shared across learners. Representations are down-projected with PCA to a dimension of 30, to ensure computational efficiency. We employ 5 diverse tabular learners. For each candidate dataset, we evaluate every model in each condition over 5 random seeds, subsampling up to 10,000 examples per run for cost-effectiveness. Our metric is AUC for classification tasks and $R^2$ for regression tasks.

**Acceptance Criteria.** To pass the curation filter, a dataset should satisfy two conditions across at least 3 out of 5 learners: (1) For *Joint Signal*, performance over the *Joint Frozen* condition should be higher than both *Unimodal Structured* and *Unimodal Unstructured* variants. This ensures that the unstructured modality is relevant, while also prevents the dataset from collapsing into a pure Natural Language Processing or Computer Vision task; and (2) For *Task-awareness* we require that the *Joint TAR* condition will improve performance over the *Joint Frozen* condition, isolating the gain from representation tuning. Figure 4 in the Appendix illustrates the protocol over concrete examples, and Appendix C provides a formal and precise definition of the acceptance criteria, and details of the curation setup.

## 3. MulTaBench

MulTaBench is composed of 40 datasets split equally between image-tabular and text-tabular while balancing between regression and classification tasks, all satisfying our

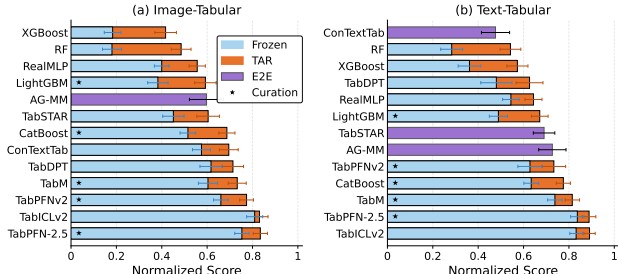

*Figure 2.* All learners gain from Target-Aware Representations.

curation pipeline established in §2. While the text-tabular subset is derived exclusively from existing benchmarks, the image-tabular subset is curated and collected from public datasets; see Appendix D for a comprehensive summary.

**Text-Tabular Curation.** To evaluate existing text-tabular benchmarks (Shi et al., 2021; Grinsztajn et al., 2023; Kim et al., 2024; Mráz et al., 2025), we aggregate all their 56 unique datasets and subject them to our 4 experimental conditions. In Figure 6 in the Appendix, we compare *Joint TAR* and *Joint Frozen* across all datasets, finding that TAR consistently outperforms frozen embeddings for all learners, highlighting the limitations of using fixed representations. With the results in hand, we apply our curation pipeline and find out that approximately 23% of the datasets fail the *Joint Signal* criterion; of the remaining datasets, 36% do not pass the *Task-awareness* criterion, leaving 41% that pass both. From these, we subsample 20 datasets to match the size of the image-tabular subset. Our acceptance rate shows that while our requirements are common enough, they do not constitute the primary focus of standard text-tabular research. Without this distinction, existing benchmarks lack the focus to research target-awareness in MMTL.

**Image-Tabular Curation.** We collect candidate datasets from existing literature (Lu et al., 2023; Tang et al., 2024b; Luo et al., 2025b; Kim et al., 2025b), identifying a shared pool of 16 unique valid datasets, from which only 5 meet our criteria (31%), a proportion comparable to the text-tabular subset. We then manually curate additional datasets from Kaggle which pass our pipeline, eventually creating the largest image-tabular benchmark to this date with 20 datasets. In the process, we encountered significant challenges, detailed in Appendix F.

## 4. Robustness Analysis

While our curation pipeline identifies datasets with high multimodal potential, it is crucial to verify that these properties remain consistent across different modeling choices.

**New Tabular Learners.** Since model ranking suffers from selection bias favoring the curation models, our ob-

jective is not to establish the SOTA, but to provide a useful tool for the development of future multimodal architectures. We supplement the original learners with 5 additional ones, while also including 3 "end-to-end" (E2E) models which natively processes texts or images. Figure 2 shows model performance on both MulTaBench subsets. Target-aware embeddings consistently outperform frozen embeddings across all new models and modalities. While this gain is expected for the curation models, its generalization to other models indicates the usefulness of our benchmark.

**Embedding Model Scale.**  So far, text and image were represented using *e5-v2-small* and *DiNO-v3-small*. Since the dimension of these embeddings is 384, one potential limitation may be that they are too small. We thus repeat the curation experiments using the *Large* variants of the models, which have approximately 10 times more parameters, and a final dimension of 1024. Figure 10 in the Appendix shows that while a larger embedding model improves downstream performance, TAR significantly outperforms frozen embeddings even at the larger scale. In fact, we even observe that the *TAR Small* variant is better than *Frozen Large*; this indicates that increased representational capacity does not guarantee that target-relevant signals are retained.

**Embedding Dimension.**  To this point, our analysis has assumed a fixed embedding size of 30 PCA components, following standard practice (Grinsztajn et al., 2023; Arazi et al., 2025). This dimensionality reduction helps prevent overfitting and ensures computational efficiency by reducing memory requirements. However, this raises the question: is TAR really surfacing information which was missing in the original representations, or is the observed gain an artifact of the compression? In Appendix H.5, we show that representation tuning remains effective across 15 and 60 dimensions, and even when removing PCA completely.

**Qualitative Analysis.**  Figure 3 and Appendix I illustrate how target-aware adaptation reshapes the encoder's focus across 4 MulTaBench datasets. In *CheXpert*, attention shifts from arbitrary anatomical borders toward the right lower lung and optic disc, respectively. Similarly, focus in *Celebs* moves from peripheral accessories to core facial features. These examples demonstrate that contextualization enables the encoder to surface specific details that are otherwise lost in task-agnostic representations.

## 5. Towards Multimodal TFMs

Our analysis of MulTaBench reveals a significant gap between current tabular learners and the demands of MMTL tasks, as existing architectures cannot jointly tune unstructured representations for the target label. In this section, we discuss the potential trajectory of future Multimodal

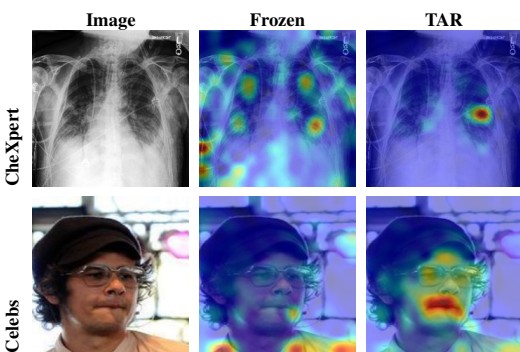

*Figure 3.* Comparison of frozen and Target-Aware Representations.

TFMs. Our vision builds upon the framework proposed by Van Breugel & Van Der Schaar (2024). Their position piece defines 4 core desiderata to guide their development: (D1) handling *mixed-type columns*, such as numbers, categories and dates, (D2) enabling *cross-dataset modeling*, (D3) leveraging *textual context* and metadata, such as column names, and (D4) maintaining *equivariance to column order*. We expand their definition by suggesting (D5) *Target-Aware Multimodal Tabular Learning*; text and image embeddings should be target-aware.

While PFNs have revolutionized structured learning, they are primarily designed for modalities where raw inputs already contain highly compressed signals. Initial efforts attempting to couple PFNs with multimodal encoders (Luo et al., 2025b; Kim et al., 2025b) have struggled to unlock TAR without violating the core ICL premise of avoiding parameter updates. In contrast, joint modeling approaches such as AutoGluon-Multimodal and TabSTAR utilize finetuning to achieve target-awareness, yet this introduces significant practical challenges. Finetuning historically complicates tabular learning by increasing overfitting risks, particularly on small-to-medium datasets, and imposing substantial computational overhead as data, model and embedding scales grow. This burden increases further when using HPO or standard practices like cross-validation and ensembling, as these methods require repeating the expensive finetuning process multiple times to find the best parameters and prevent data leakage across splits.

To summarize, we argue that none of the current architectures are optimal for MMTL, and that the leading paradigms complement each other. MulTaBench enables their development by isolating the datasets that explicitly demand task-specific representations. While proposing a solution is out of this work's scope, we believe that the optimal architecture should take the best of both worlds. An ideal model should bring the contextualization benefits of TAR while preserving the robustness and latency of ICL. We hope that the existence of MulTaBench will enable the research of such models (see Appendix J for further discussion).

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

# A. Extended Introduction

Tabular Foundation Models (TFMs) (Van Breugel & Van Der Schaar, 2024; Hollmann et al., 2022; 2025; Qu et al., 2025; Grinsztajn et al., 2026; Qu et al., 2026) have recently emerged as the state of the art (SOTA) for supervised tabular learning (Erickson et al., 2025; Ye et al., 2025). They have surpassed gradient-boosted decision trees (GBDTs) (Breiman, 2001; Chen & Guestrin, 2016; Ke et al., 2017; Prokhorenkova et al., 2018), which have historically been the leading approach (Shwartz-Ziv & Armon, 2022; Grinsztajn et al., 2022; McElfresh et al., 2023). Recently, these versatile learners have been extended to causal inference (Robertson et al., 2025), graph learning (Hayler et al., 2025), and time-series (Hoo et al., 2024). However, the best-performing TFMs (Grinsztajn et al., 2026; Qu et al., 2026) are trained exclusively on structured numerical data, making them fundamentally unimodal: unstructured inputs must be preprocessed via external embedding models (Wang et al., 2024; Siméoni et al., 2025), with no unified support for modalities such as text and image.

Yet, in many high-impact domains, tabular problems are multimodal: e-commerce listings (Das et al., 2024; Sukel et al., 2024; Shapira et al., 2024), social media feeds (Meghawat et al., 2018; Ophir et al., 2020; Badian et al.), and medical health records (Huang et al., 2020; Cui et al., 2023; Duenias et al., 2025; Fu et al., 2025) combine image and text with numerical features. While early work has begun extending TFMs to integrate text (Arazi et al., 2025; Spinaci et al., 2025), these extensions often compromise the model's core tabular performance, and inherent support for visual modalities remains entirely absent. One might turn to Large Language and Vision-Language Models (LLMs/VLMs), which natively process unstructured inputs, but they are not suited for the inductive biases of tabular data; specifically, they are unoptimized for the relational structure (Fang et al., 2024) and are suboptimal for numerical features (Van Breugel & Van Der Schaar, 2024). Addressing these limitations requires architectures that combine the numerical precision of TFMs while maintaining the rich input handling of multimodal foundation models. However, evaluating such a unified approach is difficult because the diverse nature of tasks within Multimodal Tabular Learning (MMTL) (Jiang et al., 2026; Kim et al., 2025b) is not yet fully characterized; existing benchmarks (Shi et al., 2021; Lu et al., 2023; Kim et al., 2024; Tang et al., 2024b; Mráz et al., 2025) primarily highlight the coexistence of modalities, unintentionally grouping together problems that require fundamentally different modeling solutions.

To characterize these problems, we observe that tabular models require inputs to be represented as feature columns, so high-dimensional images and texts must be compressed into compact representations. Consequently, embeddings act as lossy summaries, as they capture only a fraction of the raw input's information by design (Weller et al., 2025). In order to generalize well, pretrained embedding models are optimized for broad semantic content, such as distinguishing an X-ray from a mammogram, at the expense of fine-grained details like precise size estimations or localized anomalies (Pantazopoulos et al., 2024; Li et al., 2025). While this compression is effective for global semantic mapping, it fails to preserve the specialized signals required for fine-grained MMTL tasks. For example, the optimal representation of a chest X-ray differs depending on whether the tabular task is to diagnose pneumonia or a rib fracture, and whether the patient is a young athlete or an elderly smoker. We thus advocate for the need for Target-Aware Representations (TAR): embeddings that are tuned to the target and, ideally, to the other modalities.

Consider, for example, the task of pneumonia detection from a patient record combining age and smoking status with chest X-ray images. We argue that to study MMTL, a dataset should satisfy two properties: (1) *Joint Signal*, where each modality provides complementary information that contributes to the overall predictive performance, and (2) *Task-awareness*, where task-agnostic representations fail to capture the details required for a given objective. In our example, both the X-ray and the clinical profile offer unique, complementary information, and steering the image embedding to detect subtle signs of inflammation in the lungs should improve diagnostic accuracy.

To translate these theoretical properties into a measurable test, we develop an algorithmic pipeline that quantifies whether a dataset complies with the aforementioned requirements. This approach approximates these properties by evaluating each task across a broad suite of tabular learners, ranging from light GBDTs to SOTA TFMs. To evaluate for *Joint Signal*, we demand a performance drop when either modality is removed, verifying that each input strengthens the predictive power. For *Task-awareness*, we finetune the encoder's last 3 layers with LoRA (Hu et al., 2021) on the prediction target as a preprocessing step, and we expect these representations to outperform frozen ones when passed to tabular models. Crucially, our experiments confirm that target-aware representations outperform frozen embeddings across established MMTL benchmarks; however, we find that the magnitude of these gains is highly dataset-dependent, suggesting they represent distinct classes of MMTL tasks.

Building on this framework, we introduce **MulTaBench**, a benchmark of 40 datasets balanced between image-tabular and text-tabular tasks, as well as classification and regression objectives. To ensure a comprehensive evaluation, the benchmark

incorporates a wide range of sample sizes and feature counts, while spanning a diverse set of domains to capture the heterogeneity of real-world multimodal tabular data. MulTaBench represents the largest image-tabular benchmarking effort to date, and the first MMTL benchmark to explicitly prioritize datasets requiring task-aware representations. Demonstrating the robustness of our curation criteria, we show that the gains from target-aware tuning generalize consistently across a diverse suite of independent tabular learners, encoder scales, and embedding dimensions. These findings suggest that designing novel architectures which contextualize the representations of unstructured modalities can push the boundaries of MMTL, and we believe that MulTaBench would be instrumental for developing true Multimodal TFMs.

## B. Related Work

**Tabular Foundation Models.** The landscape of tabular learning shifted with Prior-data Fitted Networks (PFNs) (Müller et al., 2021), which pretrain transformers over synthetic tabular datasets with in-context learning (ICL) (Brown et al., 2020). The TabPFN family (Hollmann et al., 2022; 2025; Grinsztajn et al., 2026; Garg et al., 2025) pioneered this direction. Multiple subsequent works (Qu et al., 2025; 2026; Ma et al., 2025; Zhang et al., 2025a; Spinaci et al., 2025; Zhang et al., 2025b; Bouadi et al., 2025) advanced the paradigm with improvements spanning synthetic data diversity, real-world data pretraining, and architectural scalability. Among these, ConTextTab (Spinaci et al., 2025) is the only PFN to incorporate textual fields, yet it does not process raw strings; instead, it relies on external, frozen text embeddings as static inputs, decoupling the representation from the tabular learning objective. In addition, several non-PFN approaches (Yan et al., 2023; Kim et al., 2024; 2025a) also incorporate semantic awareness, but likewise treat text representations as frozen. TabSTAR (Arazi et al., 2025) represents a fundamental shift: rather than processing fixed representations, it jointly trains both the textual and tabular encoders, successfully demonstrating that TAR are essential for MMTL. However, it lacks support for images and its non-ICL architecture compromises its numerical performance.

**LLMs and VLMs.** Recent years have seen the rise of LLMs and their evolution into VLMs (Wu et al., 2023; Yin et al., 2024; Caffagni et al., 2024). These powerful models (Singh et al., 2025; Comanici et al., 2025) typically employ a unified transformer architecture (Vaswani et al., 2017) to process interleaved modalities within a single sequence, offering a path to integrate tabular data with text and image; however, research has primarily focused on text-tabular tasks (Fang et al., 2024). TabLLM (Hegselmann et al., 2023) explored different strategies to serialize the tabular data into natural language, and TabuLa-8B (Gardner et al., 2024) and TabGemma (Schindler et al., 2025) combined continued pretraining of LLMs on tabular corpora (Eggert et al., 2023) with architectural modifications, achieving strong few-shot performance. Nevertheless, the autoregressive nature of LLMs is misaligned with the structure of tabular data, and their tokenization process damages numerical precision (Thawani et al., 2021; Spathis & Kawsar, 2024). Furthermore, their massive scale introduces prohibitive costs for high-throughput inference, while their extensive pretraining risks memorizing evaluation data (Bordt et al., 2024; Gorla & Puduppully, 2026). Consequently, generative architectures remain largely impractical for discriminative MMTL.

**Joint Multimodal Tabular Learning Architectures.** Despite various architectural proposals (Hager et al., 2023; Jiang et al., 2024; Ebrahimi et al., 2024; Hu et al., 2024; Du et al., 2025), the field still lacks a true multimodal foundation model for tabular data with text and images. AutoML (He et al., 2021) frameworks (Shi et al., 2021; Tang et al., 2024a;b), led by AutoGluon-Multimodal (Tang et al., 2024a), demonstrated the benefit of joint modeling by combining tabular, text and image encoders. However, their reliance on a non-ICL transformer (Gorishniy et al., 2021) as the tabular backbone limits their tabular capacities. Similarly, TabSTAR (Arazi et al., 2025) introduced a jointly pretrained text-tabular architecture and achieved strong performance on text-tabular classification tasks, but it struggled with regression tasks and with unimodal tabular benchmarks (Erickson et al., 2025). Recent attempts have built on stronger tabular foundations, by expanding the PFN paradigm with multimodal fusion strategies. TIME (Luo et al., 2025b) proposed a late-fusion approach in an image-tabular setup, but missed cross-modal interactions and achieved mixed results when employing finetuning. MultiModalPFN (Kim et al., 2025b) fused TabPFN with visual and textual backbones, but assumed frozen multimodal embeddings. To conclude, no existing model has successfully maintained SOTA performance on tabular tasks while learning TAR for text and images.

**Text-Tabular Benchmarks.** Existing text-tabular benchmarks differ significantly in their curation philosophy and dataset scale. The Multimodal AutoML Benchmark (Shi et al., 2021) introduced 18 datasets with deliberate diversity in task type and predictive signal. Grinsztajn et al. (2023) filtered 14 datasets from a bigger pool, where the text features provided a significant gain over a numerical-only baseline. TextTabBench (Mráz et al., 2025) curated 13 text-tabular datasets, focusing on longer text fields while ensuring both the text modality and numerical features contribute to the prediction. CARTE (Kim et al., 2024) collected 51 datasets, mainly featuring short strings and high-cardinality categories, typically present

in knowledge graphs. While these efforts were instrumental in advancing research on tabular data with strings, none of them were deliberately designed to isolate tasks where static representations fail to capture the necessary predictive signal. Importantly, as we show in § 3, most of the datasets included in the aforementioned benchmarks do not pass our curation pipeline. Consequently, potential performance gains that native Multimodal TFMs are designed to deliver might be overlooked. For example, ConTextTab set the SOTA for the CARTE benchmark (Spinaci et al., 2025), but struggles on MulTaBench (see § 4).

**Image-Tabular Benchmarks.** The availability of image-tabular benchmarks is highly limited. MuG (Lu et al., 2023) introduced 4 data sources from the gaming domain combining tabular data with text and image, but offering limited domain diversity. Similarly, Tang et al. (2024b) curated 11 tabular datasets with images, but without quantifying the image signal's necessity. As detailed in § 3, these datasets often fail our curation pipeline and suffer from additional quality issues. The lack of large accessible benchmarks led recent work such as TIME (Luo et al., 2025b) and MultimodalTabPFN (Kim et al., 2025b), to rely on a self-selected group of datasets, limiting the generalizability of their findings. We address this gap by doubling the benchmark size and assuring that the image representations are central for MMTL.

**Limits of Frozen Representations.** Pretrained representations are optimized for general-purpose objectives and often fail to capture the fine-grained, task-specific details necessary for downstream performance (Tong et al., 2024; Liu et al., 2025; Gisserot-Boukhlef et al., 2025; Cao et al., 2026). Weller et al. (2025) provide a theoretical basis for this limitation, demonstrating how RAG systems (Lewis et al.) that rely on static embeddings can fail on even seemingly simple cases. To overcome this problem, alternative approaches (Khattab & Zaharia, 2020; Malaviya et al., 2023; Fan et al., 2024; Tang & Yang, 2024; Edge et al., 2025; Wang et al., 2025; Pu et al., 2025; Koshorek et al., 2025) enabled the contextualization of document representations in the presence of the query. Similar limitations were also illustrated in VQA (Antol et al., 2015), where encoding images independently of the question leads to information loss, as the query determines which image regions are predictive (Ganz et al., 2024; Li et al., 2025). To overcome these limitations, VLMs have evolved toward deep multimodal alignment (Radford et al., 2021; Alayrac et al., 2022; Liu et al., 2023), and we argue that MMTL should undergo a similar evolution, moving away from decoupled preprocessing and frozen embeddings in favor of a joint learning approach.

# C. Curation Pipeline

## C.1. Target-Aware Representations

Target-Aware Representations are produced by finetuning the top 3 transformer layers of the encoder using LoRA (Hu et al., 2021), with a single linear head mapping the encoder output (384-dim) to the number of output classes. Finetuning is performed as a preprocessing step, independently of the structured features and the downstream tabular learner. The encoder is adapted on the training split only, using a stratified 90/10 train/validation split to select the best checkpoint. Importantly, there is no data leakage, as the test set is never used for this step, just like any other preprocessing.

**Hyperparameters.** Both DINO-v3-small[1] and e5-small-v2[2] share the same LoRA configuration: $r = 16$, $\alpha = 32$, dropout 0.1. Training uses AdamW with learning rate $10^{-4}$ for e5 and 0.001 for DINO, with a batch size of 256, and weight decay 0.01. For DINO, we train to up to 100 epochs. As many datasets have multiple text features, we reduce this number for e5 to 50. We apply early stopping after 3 epochs of no improvement on the validation loss. All hyperparameters are fixed across datasets; no per-dataset tuning is performed. Reported gains are therefore conservative lower bounds on what task-specific adaptation could achieve.

**Regression.** For regression targets, the continuous label is discretized into 20 equal-frequency bins and the adaptation objective is cross-entropy over these bins. We find this technique to be more stable than direct regression finetuning, as it is much less sensitive to outliers. However, it's plausible that this decision could be optimized much further.

**Text.** While MulTaBench image datasets have a single image feature, text-tabular datasets often have more than one text field, which we defined as string features that have at least 100 distinct values. For efficiency, a single e5 model is finetuned jointly across all text columns: each row-col pair generates one training example in the format "$col\_name : col\_val$", paired

---

[1] https://huggingface.co/facebook/dinov3-vits16-pretrain-lvd1689m
[2] https://huggingface.co/intfloat/e5-small-v2

with the row's target label. This allows the model to learn a shared representation across all text features simultaneously. This decision might harm representations, especially as feature size grows, but finetuning a dedicated embedding model for each feature would have been computationally infeasible.

## C.2. Curation Experimental Setup

Each candidate dataset is evaluated by 5 tabular learners: LightGBM, CatBoost, TabM, TabPFNv2, and TabPFN-2.5, over five random seeds under the 4 conditions defined in §2.2. Training is capped at 10,000 examples per fold, and the metric is AUC for classification and $R^2$ for regression tasks.

We run models using default configurations. For LightGBM, we use its default implementation[3]. For CatBoost, we follow previous work (Gorishniy et al., 2021; Arazi et al., 2025) and set $early\_stopping\_rounds = 50, od\_pval = 0.001, iterations = 2000$. For TabM, we use its *pytabkit*[4] implementation with default parameters. For TabPFNv2 and TabPFN-2.5, we use their default implementation.[5]

*Table 1.* Experimental Conditions. Breakdown by feature composition and representation strategy.

| Condition | Structured | Unstructured | Target-Aware (TAR) |
|---|---|---|---|
| Unimodal Structured | ✓ | ✗ | − |
| Unimodal Unstructured | ✗ | ✓ | ✗ |
| Joint Frozen | ✓ | ✓ | ✗ |
| Joint TAR | ✓ | ✓ | ✓ |

## C.3. Formal Acceptance Criteria

Let $\mathcal{D}$ be a candidate dataset and $\mathcal{M}$ be a pool of 5 curation tabular learners. For a given learner $m \in \mathcal{M}$, let $S_m(\text{Condition})$ denote its average predictive performance (AUC or $R^2$) under a given condition.

**Joint Signal.** We define the *Joint* gain as the improvement of the joint model over the strongest unimodal baseline:

$$\Delta_{\text{Joint}}(m) = S_m(\text{Joint Frozen}) - \max\big(S_m(\text{UnimodalStructured}), \ S_m(\text{UnimodalUnstructured})\big)$$

**Task-awareness.** We define the *Awareness* gain as the improvement of Joint TAR over Joint Frozen:

$$\Delta_{\text{Awareness}}(m) = S_m(\text{Joint TAR}) - S_m(\text{Joint Frozen})$$

**Selection rule.** To ensure that the observed improvements are robust and exceed a minimum significance margin, we introduce a threshold parameter $\delta \geq 0$ and a consensus fraction $\rho \in (0.5, 1]$. A dataset $\mathcal{D}$ is accepted if and only if both gains exceed $\delta$ for a majority of the learners:

$$\text{Accept}(\mathcal{D}) \iff |\{m \in \mathcal{M} : \Delta_{\text{Joint}}(m) > \delta \ \wedge \ \Delta_{\text{Awareness}}(m) > \delta\}| \geq \rho \cdot |\mathcal{M}|$$

The two conditions are evaluated jointly per learner: a model counts toward the threshold only if both gains are above the threshold. In our case, we set $\delta = 0.001$ and $\rho = 3/5$. We note that since we use a binary decision threshold, some datasets can marginally cross it while others can generate consensus. Demanding stricter thresholds could enhance the robustness of the selected datasets.

---

[3]https://pypi.org/project/lightgbm/
[4]https://github.com/dholzmueller/pytabkit
[5]https://github.com/PriorLabs/TabPFN

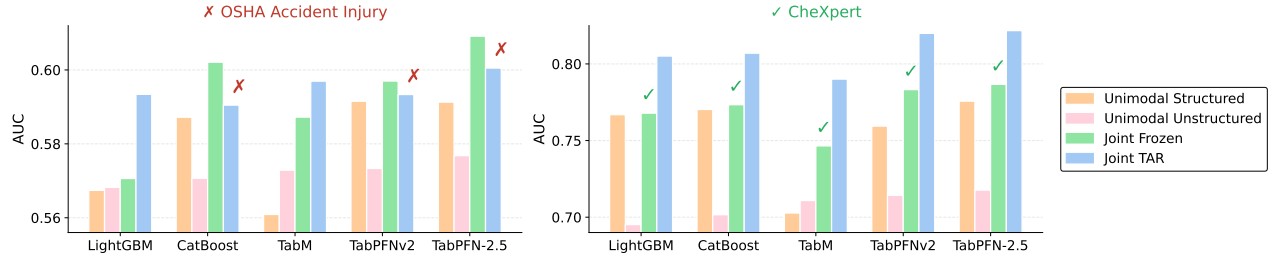

*Figure 4.* Curation protocol over candidate datasets. Mean AUC per model and condition. The *OSHA Accident Injury* dataset is rejected as *TAR* fails to consistently improve over *Joint Frozen*.

# D. MulTaBench Datasets

In this section we present MulTaBench datasets. Table 2 provides their high-level statistics, including the number of rows and feature type breakdown. The rest of the section provides a concise per-dataset high-level description; their exact preprocessing logic can be found in our released code.

## D.1. Image-Tabular Dataset Descriptions

**CBIS-DDSM.**   Cropped mammography mass regions from the Curated Breast Imaging Subset of DDSM, with 1,696 crops. The 4-class target is BI-RADS breast density (categories 1–4). Structured features describe lesion morphology, such as laterality, imaging view (MLO or CC), mass shape, mass margins, BI-RADS assessment score, pathology (malignant/benign), and subtlety rating.

**Celeb Attractiveness.**   Celebrity face images from the CelebA dataset, sampled to 99,999 images[6] from the full 202,599. The binary target is a crowd-annotated attractiveness label. Each row pairs the face image with 39 binary facial-attribute features, such as *Smiling*, *Wearing_Lipstick* or, *Bald*, making the image a complement to an already rich structured signal.

**CheXpert.**   Chest X-ray images from the Stanford CheXpert dataset, with 46,437 frontal and lateral views. The 3-class target predicts Cardiomegaly label (positive, negative, or uncertain). Structured features include patient sex, age, and 14 co-occurring pathology labels, many of which are sparsely observed (over 85% missing for several conditions), reflecting natural label uncertainty in radiology reports.

**CS:GO Skins.**   Weapon skin images and metadata from the Counter-Strike: Global Offensive marketplace, with 956 cosmetic items. The 10-class target discretizes market price into decile quantile bins. Structured features include skin quality (rarity tier), weapon category, and availability; a free-text skin name column provides additional descriptive signal about the skin's visual design.

**Flower Bouquets.**   Flower bouquet photographs paired with sales metadata from a Russian online florist, comprising 600 listings. The 5-class target is a customer satisfaction rating (1–5). Features include a free-text bouquet description, average comment-based rating, and price.

**Glaucoma SMDG.**   Retinal fundus photographs from the SMDG multi-source glaucoma benchmark, with 12,449 images. The 3-class target encodes glaucoma diagnosis (positive, negative, or uncertain).Clinical metadata including patient age, sex, laterality, and intraocular pressure are available as structured features, though heavily sparse (over 99% missing for several fields), reflecting real-world incompleteness in ophthalmic records.

**Hateful Meme.**   Multimodal memes from the Facebook Hateful Memes Challenge, comprising 10,000 image-text pairs. The binary target labels each meme as hateful or not. To make it a tabular task, we pre-embedded the text field into 20 continuous variables, to capture part (but not all) of the text signal. The structured columns should thus be treated as numeric features rather than raw text, with the meme image providing complementary visual context.

---

[6]This was originally intended to be 100,000, but we eventually dropped an observation with a corrupted image.

*Table 2.* All 40 MulTaBench Datasets Properties. *Task*: Classification (CLS) or Regression (REG). *Classes*: number of target classes (for CLS). *N*: total examples. *Struct.*: numerical + categorical features. *Text*: free-text features. *Img.*: image features.

| Dataset | Task | Classes | $N$ | Struct. | Text | Img. |
|---|---|---|---|---|---|---|
| *Image-Tabular (20 datasets)* | | | | | | |
| CBIS-DDSM | CLS | 4 | 1,696 | 8 | 0 | 1 |
| Celeb Attractiveness | CLS | 2 | 99,999 | 39 | 0 | 1 |
| CheXpert | CLS | 3 | 46,437 | 17 | 0 | 1 |
| CS:GO Skins | CLS | 10 | 956 | 3 | 1 | 1 |
| Flower Bouquets | CLS | 5 | 600 | 3 | 1 | 1 |
| Glaucoma SMDG | CLS | 3 | 12,449 | 8 | 0 | 1 |
| Hateful Meme | CLS | 2 | 10,000 | 20 | 0 | 1 |
| HubMAP HPA | CLS | 10 | 12,581 | 3 | 1 | 1 |
| Justin Instagram | CLS | 5 | 10,319 | 6 | 0 | 1 |
| Mammography CMMD | CLS | 2 | 5,202 | 4 | 0 | 1 |
| PetFinder | CLS | 8 | 14,652 | 17 | 4 | 1 |
| Zooscan Plankton | CLS | 10 | 100,000 | 28 | 0 | 1 |
| Amazon Bestseller | REG | – | 3,488 | 4 | 0 | 1 |
| Amazon Packages | REG | – | 46,398 | 1 | 1 | 1 |
| H&M Fashion | REG | – | 104,072 | 9 | 4 | 1 |
| Khaadi Clothes | REG | – | 400 | 2 | 1 | 1 |
| Letterboxd Movies | REG | – | 12,564 | 23 | 3 | 1 |
| Mango Mass | REG | – | 546 | 2 | 0 | 1 |
| MkPhoto Bots | REG | – | 13,748 | 8 | 0 | 1 |
| Painting Price | REG | – | 12,369 | 245 | 2 | 1 |
| *Text-Tabular (20 datasets)* | | | | | | |
| Data Scientist Salary | CLS | 6 | 15,841 | 1 | 5 | 0 |
| Fake Job Postings | CLS | 2 | 12,725 | 2 | 3 | 0 |
| Jigsaw Toxicity | CLS | 2 | 100,000 | 29 | 2 | 0 |
| Kickstarter | CLS | 2 | 86,502 | 4 | 5 | 0 |
| Michelin Guide | CLS | 5 | 18,843 | 5 | 6 | 0 |
| Product Sentiment | CLS | 4 | 5,091 | 1 | 1 | 0 |
| Spotify Genres | CLS | 114 | 114,000 | 15 | 3 | 0 |
| US Accidents | CLS | 4 | 100,001 | 35 | 9 | 0 |
| Wine Review | CLS | 30 | 84,123 | 3 | 2 | 0 |
| Women's Clothing | CLS | 5 | 18,788 | 8 | 2 | 0 |
| Baby Products | REG | – | 5,085 | 8 | 4 | 0 |
| Book Price | REG | – | 4,989 | 3 | 5 | 0 |
| Book Readability | REG | – | 4,724 | 24 | 6 | 0 |
| Mercari Marketplace | REG | – | 100,000 | 3 | 6 | 0 |
| Montgomery Salaries | REG | – | 9,228 | 7 | 4 | 0 |
| Rotten Tomatoes | REG | – | 7,158 | 2 | 13 | 0 |
| SciMagojr Impact | REG | – | 31,136 | 12 | 10 | 0 |
| Vancouver Salaries | REG | – | 44,574 | 3 | 2 | 0 |
| Video Games Sales | REG | – | 16,598 | 3 | 2 | 0 |
| Zomato Restaurants | REG | – | 41,665 | 8 | 7 | 0 |

**HubMAP HPA.** Histology tissue tile images from the HuBMAP-HPA organ segmentation competition, with 12,581 tiles. The 10-class target discretizes donor age into decile quantile bins, asking whether tissue morphology encodes biological age. Structured features include organ type (kidney, prostate, large intestine, spleen, lung), donor sex, and tile coordinates; a run-length encoding column of the segmentation mask is present but largely absent (61% missing).

**Justin Instagram.** Instagram posts from five celebrities named Justin (Bieber, Trudeau, Timberlake, Long, Hartley), totaling 10,319 posts. The 5-class target identifies which Justin authored each post. Structured features are post-level metadata: number of hashtags, characters, words, emojis, and mentions, plus a binary video indicator.

**Mammography CMMD.** Mammography images from the Chinese Mammography Database, with 5,202 cropped lesion regions. The binary target distinguishes malignant from benign findings. Structured features include patient age, laterality (left/right), abnormality type (mass, calcification, or both), and the cropping method used (YOLO or contour detection).

**PetFinder.** Pet adoption listings from the Malaysian PetFinder platform, with 14,652 entries. The 8-class target discretizes listed pet age into octile bins, testing whether visual appearance and listing text jointly predict developmental stage. Features include species (cat/dog), breed, color, health status (vaccinated, dewormed, sterilized), adoption fee, state location, and a free-text listing description alongside the pet's photograph.

**Zooscan Plankton.** Underwater zooplankton specimens from the PELGAS Bay of Biscay survey, scanned with a ZooScan optical system, totaling 100,000 specimens. The 10-class target classifies copepod taxa (Calanoida, Oithonidae, Calanidae, Temoridae, and others). Structured features include 28 morphometric descriptors computed from the scan (circularity, skewness, fractal dimension, symmetry scores, area coverage, etc.) alongside sampling metadata such as geographic coordinates, depth, collection date, and mesh size.

**Amazon Bestseller.** Product listings from Amazon's bestseller rankings across all departments, with 3,488 items. The target is log-transformed product price. Structured features are the number of ratings, bestseller rank within department, star rating, and list page; the product thumbnail image provides visual cues about item type and packaging.

**Amazon Packages.** Warehouse bin images from Amazon's robotic fulfillment centers, with 46,398 bins. The target is the total weight of the bin's contents in pounds. The sole structured feature is the expected item count; a free-text product description column names the item in each bin.

**H&M Fashion.** Clothing article metadata and thumbnail images from H&M's product catalog, with 104,072 articles. The target is the average age of purchasing customers, capturing whether visual style and descriptive text encode demographic appeal. Structured attributes include product type, color group, graphical appearance, garment group, and department; text features are the product name and a free-text detail description, making this a trimodal dataset.

**Khaadi Clothes.** Apparel listings from the Pakistani fashion brand Khaadi, with 400 products. The target is retail price in Pakistani rupees. Structured features are color and product category; a free-text description column specifies fabric type and construction.

**Letterboxd Movies.** Film metadata and poster images from the Letterboxd movie-tracking platform, with 12,564 films released between 2021 and 2024. The target is the average community rating. Features include 19 binary genre flags, release year, runtime, and text fields for movie tagline and theme descriptions alongside the official poster image.

**Mango Mass.** Mango fruit photographs from a variety classification study, with 546 individual fruits. The target is fruit mass in kilograms. The only structured features are color group (yellow or green) and quality grade (1, 2, or premium), making the image the dominant signal for weight prediction.

**MkPhoto Bots.** Social media photographs collected for image authenticity research, with 13,748 posts. The target is a continuous trust score reflecting the estimated probability that the post is genuine. Structured features include binary flags for GAN generation and deepfake manipulation, presence of a person, face count, a recognized-celebrity list, upload speed, and a noise-quality score.

**Painting Price.**    Painting images and metadata from an online art marketplace, with 12,369 works. The target is sale price. Structured features include physical dimensions (width, length), material (canvas, paper, wood, etc.), and 243 binary style tags (e.g., *abstract*, *impressionism*, *surrealism*); a high-cardinality free-text styles column provides additional stylistic signal.

## D.2. Text-Tabular Dataset Descriptions

**Data Scientist Salary.**    Indian data science job postings, with 15,841 listings. The 6-class target is salary band in lakh rupees per annum (0–3, 3–6, 6–10, 10–15, 15–25, 25–50). Text features include the experience range, job description (22% missing), job designation, required key skills, and city location; a noisy job-type field (75% missing) contributes as a weak structured signal.

**Fake Job Postings.**    Job listings annotated for authenticity, with 12,725 postings. The binary target flags fraudulent listings. Text features include the job title and full description; structured features capture required experience level, required education, and salary range (83% missing), testing whether deceptive intent is expressed in free-text beyond coarse metadata.

**Jigsaw Toxicity.**    Online comments from the Civil Comments platform, collected for Jigsaw's toxicity detection task, sampled to 100,000 instances. The binary target labels each comment as toxic. Alongside the comment text, structured features include 24 identity-mention fraction scores (e.g., proportions of annotators who identified references to religion, race, or gender; 77.5% missing) and five community-reaction counts (funny, wow, sad, likes, disagree).

**Kickstarter.**    Crowdfunding campaigns from Kickstarter, with 86,502 projects. The binary target indicates whether the funding goal was reached. Text features are the project name, description, and keyword slug; structured features include the funding goal amount, country, currency, and campaign deadline and creation timestamps.

**Michelin Guide.**    Restaurant listings from the 2021 Michelin Guide, with 18,843 restaurants worldwide. The 5-class target is the Michelin award level: Selected Restaurants, Bib Gourmand, and 1–3 Stars. Text features include restaurant name, address, city/country location, cuisine type, facilities and services, and a detailed Michelin editorial description; structured features are geo-coordinates, price tier, and a Green Star sustainability flag.

**Product Sentiment.**    Tweets about Apple, Google, and Twitter products posted during SXSW 2011, with 5,091 posts. The 4-class target is sentiment: Positive, Negative, No Sentiment, or Cannot Say. The sole text feature is the tweet content; a numeric product-type column (10 integer-encoded product categories) identifies the product being discussed.

**Spotify Genres.**    Spotify track metadata covering 1,000 tracks per genre across 114 genres, totaling 114,000 tracks. The 114-class target is the track genre. Text features include artist name, album name, and track name; structured features are 15 Spotify audio descriptors (danceability, energy, loudness, speechiness, acousticness, instrumentalness, liveness, valence, tempo, and others).

**US Accidents.**    Traffic accident records from the contiguous United States, sampled to 100,001 incidents. The 4-class target is accident severity on a 1–4 scale. Text features include a free-text incident description and eight location and weather text columns (street name, city, county, state, ZIP code, nearest airport code, weather condition, wind direction); structured features cover GPS coordinates, weather measurements, timestamps, and 12 binary road-feature flags.

**Wine Review.**    Professional wine tasting notes from Wine Enthusiast magazine, with 84,123 reviews. The 30-class target is the grape variety. Text features are the tasting note description and province of origin; structured features are the numeric rating (points), price (6.6% missing), and country, making grape identification from flavor language a natural benchmark for text-tabular models.

**Women's Clothing.**    Customer reviews of women's clothing from an anonymous US e-commerce retailer, with 18,788 reviews. The 5-class target is the star rating (1–5). Text features are the review title and full review text; structured features include customer age, product and department metadata, a binary recommendation indicator, and positive feedback count.

**Baby Products.**    Nursery and baby product listings from a US retail catalog, with 5,085 items. The target is retail price. Text features are the product title, free-form brand name, and descriptive fields for color, fabric, and material (all sparsely

populated at 50–99% missing); structured features include a discount flag, product category, and physical dimensions (weight, length, width, height).

**Book Price.**   Books listed on an online marketplace, with 4,989 titles. The target is log-transformed price in USD. Text features include the book title, author name, edition details, full synopsis, genre tag, and broad book category; structured features are average star rating and number of ratings (16.7% missing).

**Book Readability.**   Text excerpts from the CLEAR Corpus, with 4,724 passages from children's and educational literature. The target is the New Dale–Chall Readability Formula score, a standard measure of text difficulty. The key text feature is the excerpt itself; structured features comprise 24 linguistic and bibliographic attributes including publication year, sentence and paragraph count, Flesch–Kincaid grade level, ARI, SMOG, and CAREC readability metrics, MPAA content rating, and Bradley–Terry easiness score.

**Mercari Marketplace.**   Secondhand item listings from the Mercari mobile marketplace, sampled to 100,000 listings. The target is log-transformed sale price. Text features are the item name, free-text item description, and a three-level hierarchical category label; structured features are item condition (1–5), brand name (42.5% missing), and a binary shipping-included flag.

**Montgomery Salaries.**   Annual salary records of Montgomery County (Maryland, USA) government employees, with 9,228 employees. The target is current annual salary. Text features include department name, division, job title, and underfilled title (88.2% missing); structured features are gender, 2016 gross pay, 2016 overtime pay (31.6% missing), assignment type (full/part time), and hire date.

**Rotten Tomatoes.**   Movie metadata aggregated from IMDb and Rotten Tomatoes, with 7,158 films. The target is an audience/critic composite rating. Text features include movie name, director, screenwriter, full cast list, language, country, filming locations, genre tags, and plot description; structured features are release year, runtime, and rating and review counts.

**SciMagojr Impact.**   Academic journal and book series metadata from the SCImago Journal & Country Rank database, with 31,136 entries. The target is the journal's H-index. Text features include the journal title, publisher name, coverage period, subject categories, and broad subject areas; structured features are the SJR impact score, quartile ranking, annual and three-year document and citation counts, Overton policy citation index, and SDG alignment score.

**Vancouver Salaries.**   Annual salary disclosures for City of Vancouver public employees, with 44,574 records spanning 2007–2024. The target is annual remuneration. Text features are job title and department name; structured features are fiscal year, employee name, and declared expenses (5.4% missing).

**Video Games Sales.**   Video game sales records from VGChartz, with 16,598 titles. The target is global sales in millions of units. Text features are game title and publisher name; structured features are platform (31 gaming systems), release year, and genre (12 categories).

**Zomato Restaurants.**   Restaurant listings from the Zomato platform covering Bangalore, India, with 41,665 restaurants. The target is the aggregate user rating (ranging from 3.3 to 4.2). Text features include restaurant name, address, cuisine types, customer-highlighted dishes, raw user review text, and menu item lists; structured features include online ordering and table reservation availability, total votes, neighborhood location, restaurant type, and approximate cost for two.

## E. Text-Tabular Curation

We evaluate existing text-tabular benchmarks by drawing candidates from 4 sources: the Multimodal AutoML Benchmark (Shi et al., 2021), Grinsztajn et al. (2023), CARTE (Kim et al., 2024), and TextTabBench (Mráz et al., 2025), yielding 56 unique candidates after deduplication and exclusion of datasets which were unavailable due to improper hosting. Each dataset is evaluated by 5 tabular learners over 5 folds under the 4 conditions defined in §2.2.

### E.1. Existing Benchmarks

The 4 source benchmarks share a substantial number of datasets, either by directly using the exact same source or by using similar-enough datasets. We adopt the deduplication performed by Arazi et al. (2025), and extend it to include TextTabBench (Mráz et al., 2025), yielding a pool of 56 unique datasets. Table 3 shows datasets which are shared across more than one existing text-tabular benchmark.

*Table 3.* Duplicate datasets across benchmarks. ✓ indicates presence.

| Dataset | AutoML Multimodal | Grinsztajn et al | CARTE | TextTabBench |
|---|---|---|---|---|
| Wine Reviews | ✓ | ✓ | ✓ | ✓ |
| Zomato Restaurants | | ✓ | ✓ | |
| Vancouver Salaries | | ✓ | ✓ | |
| Company Employees | | ✓ | ✓ | |
| Montgomery Salaries | | ✓ | ✓ | |
| Ramen Ratings | | ✓ | ✓ | |
| Bike Bikewale | | ✓ | ✓ | |
| Book Readability | | ✓ | ✓ | |
| US Accidents | | ✓ | ✓ | |
| Mercari Marketplace | ✓ | | | ✓ |
| California House Prices | ✓ | | | ✓ |
| Kickstarter Funding | ✓ | | | ✓ |
| Fake Job Posting | ✓ | | | ✓ |
| Spotify Genres | | ✓ | | ✓ |
| Beer Ratings | | | ✓ | ✓ |

### E.2. Empirical Results for Curation Conditions

Figure 5 shows normalized scores across all 4 conditions for the full pool and the MulTaBench subset. The *Structured* and *Unstructured* bars serve as unimodal baselines. The MulTaBench subset shows a consistent ordering across all 4 conditions, which is more pronounced than in the full corpus.

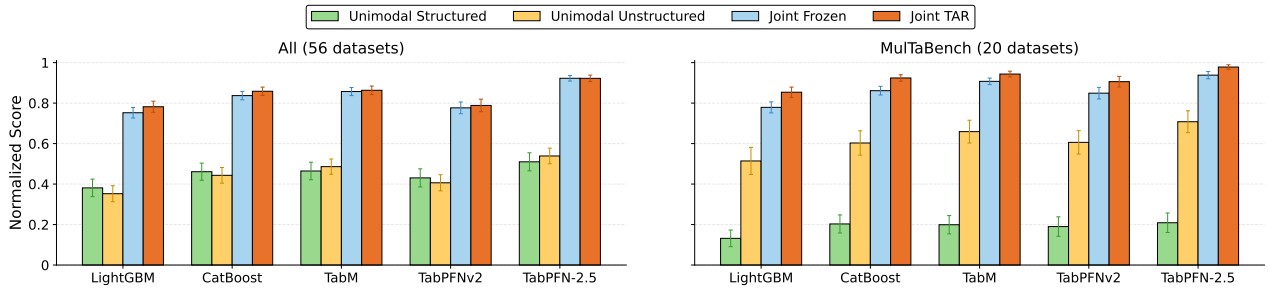

*Figure 5.* Curations Conditions for the Text-Tabular Pool. Normalized scores for *Structured*, *Unstructured*, *Joint Frozen*, and *Joint TAR* across all 56 candidates (left) and the MulTaBench subset (right).

### E.3. Benchmark Acceptance Breakdown

Table 4 reports acceptance rates per source benchmark. Grinsztajn et al. and the AutoML Multimodal Benchmark yield the highest rates. CARTE has the lowest acceptance rate (33%), reflecting its focus on knowledge-graph-style short strings and high-cardinality categorical columns. Out of 56 candidates, 23 pass all criteria; we retain 20 for MulTaBench.

### E.4. Per-Dataset Curation Results

Table 5 reports, for each of the 56 candidate datasets, whether each of the five curation models satisfies both criteria jointly. A checkmark indicates both hold simultaneously; × indicates failure on at least one; − indicates the model could not be evaluated, due to highly-multiclass problems where TabPFN's variants can't run on. Datasets are sorted approved-first, then by descending pass count.

*Table 4.* Text-tabular curation acceptance rates by source benchmark.

| Benchmark | Candidates | Accepted | Rate |
|---|---|---|---|
| AutoML Multimodal | 16 | 10 | 62% |
| Grinsztajn et al. | 11 | 7 | 64% |
| CARTE | 33 | 11 | 33% |
| TextTabBench | 13 | 7 | 54% |
| **Total (deduplicated)** | **56** | **23** | **41%** |

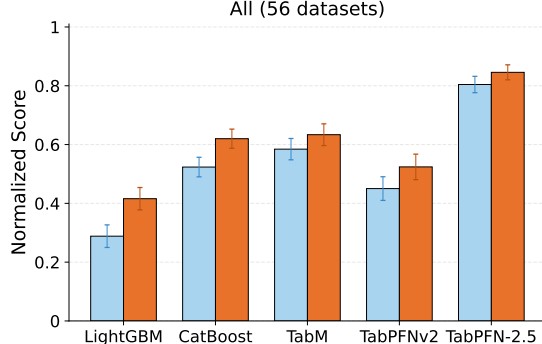
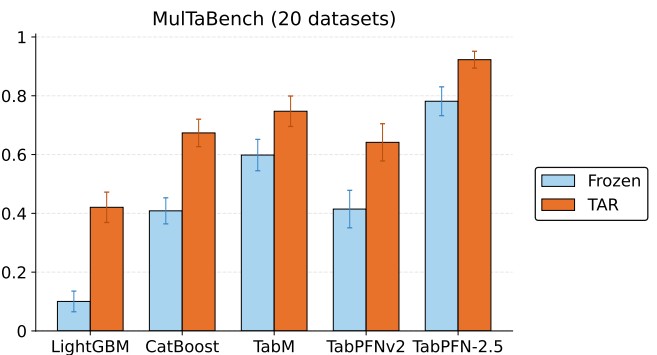

*Figure 6.* Target-Aware Representations Gains over Frozen. Normalized scores for *Joint TAR* and *Joint Frozen* across all text-tabular benchmark datasets (left), and its MulTaBench subset (right).

*Table 5.* Per-dataset curation grid. Models: LightGBM (LGBM), CatBoost (Cat), TabM, PFNv2 (TabPFNv2), PFN-2.5 (TabPFN-2.5). Each cell indicates whether the model satisfies criteria. *Pass?* column shows how many models pass. Options are pass (✓), fail (×), and N/A (−).

| Dataset | LGBM | Cat | TabM | PFNv2 | PFN-2.5 | Pass? |
|---|---|---|---|---|---|---|
| *Approved (23 datasets)* | | | | | | |
| Kickstarter Funding | ✓ | ✓ | ✓ | ✓ | ✓ | 5 |
| Jigsaw Toxicity | ✓ | ✓ | ✓ | ✓ | ✓ | 5 |
| Product Sentiment | ✓ | ✓ | ✓ | ✓ | ✓ | 5 |
| Women's Clothing | ✓ | ✓ | ✓ | ✓ | ✓ | 5 |
| Michelin Guide | ✓ | ✓ | ✓ | ✓ | ✓ | 5 |
| News Channel Category | ✓ | ✓ | ✓ | ✓ | ✓ | 5 |
| Baby Products | ✓ | ✓ | ✓ | ✓ | ✓ | 5 |
| Vancouver Salaries | ✓ | ✓ | ✓ | ✓ | ✓ | 5 |
| SciMagojr Impact | ✓ | ✓ | ✓ | ✓ | ✓ | 5 |
| Book Readability | ✓ | ✓ | ✓ | ✓ | ✓ | 5 |
| Video Games Sales | ✓ | ✓ | ✓ | ✓ | ✓ | 5 |
| Consumer Complaint | ✓ | ✓ | ✓ | ✓ | × | 4 |
| Hearthstone Cards | ✓ | ✓ | × | ✓ | ✓ | 4 |
| US Accidents | ✓ | ✓ | × | ✓ | ✓ | 4 |
| Book Price | ✓ | ✓ | ✓ | ✓ | × | 4 |
| Mercari Marketplace | ✓ | ✓ | ✓ | ✓ | × | 4 |
| Zomato Restaurants | ✓ | ✓ | ✓ | ✓ | × | 4 |
| Rotten Tomatoes | ✓ | ✓ | × | ✓ | ✓ | 4 |
| Fake Job Posting | ✓ | ✓ | × | ✓ | × | 3 |
| Wine Review | ✓ | ✓ | ✓ | − | − | 3 |
| Data Scientist Salary | × | ✓ | × | ✓ | ✓ | 3 |
| Spotify Genres | ✓ | ✓ | ✓ | − | − | 3 |

*(continued from previous page)*

| Dataset | LGBM | Cat | TabM | PFNv2 | PFN-2.5 | Pass? |
|---|---|---|---|---|---|---|
| Montgomery Salaries | ✕ | ✕ | ✓ | ✓ | ✓ | 3 |
| *Rejected (33 datasets)* | | | | | | |
| OSHA Accident Injury | ✓ | ✕ | ✓ | ✕ | ✕ | 2 |
| Google Q&A Type | ✕ | ✕ | ✕ | ✓ | ✓ | 2 |
| American Eagle Prices | ✕ | ✓ | ✓ | ✕ | ✕ | 2 |
| JC Penney Products | ✓ | ✕ | ✕ | ✓ | ✕ | 2 |
| Wikiliq Alcohol | ✕ | ✕ | ✓ | ✓ | ✕ | 2 |
| Chocolate Bar Ratings | ✕ | ✓ | ✓ | ✕ | ✕ | 2 |
| Wine Vivino Spain | ✓ | ✕ | ✓ | ✕ | ✕ | 2 |
| California House Prices | ✓ | ✓ | ✕ | ✕ | ✕ | 2 |
| SF Permit Applications | ✕ | ✕ | ✓ | ✓ | ✕ | 2 |
| FIFA22 Wages | ✓ | ✕ | ✕ | ✓ | ✕ | 2 |
| IMDB Genre | ✕ | ✕ | ✕ | ✓ | ✕ | 1 |
| Melbourne Airbnb | ✓ | ✕ | ✕ | ✕ | ✕ | 1 |
| Bike Price Bikewale | ✕ | ✕ | ✕ | ✓ | ✕ | 1 |
| Car Price Cardekho | ✓ | ✕ | ✕ | ✕ | ✕ | 1 |
| Polish Wine Prices | ✕ | ✓ | ✕ | ✕ | ✕ | 1 |
| ML/DS Job Salaries | ✕ | ✕ | ✓ | ✕ | ✕ | 1 |
| Books Goodreads | ✕ | ✓ | ✕ | ✕ | ✕ | 1 |
| Korean Drama | ✓ | ✕ | ✕ | ✕ | ✕ | 1 |
| US Museum Revenues | ✓ | ✕ | ✕ | ✕ | ✕ | 1 |
| Used Cars Pakistan | ✓ | ✕ | ✕ | ✕ | ✕ | 1 |
| Used Cars Saudi Arabia | ✕ | ✕ | ✕ | ✕ | ✓ | 1 |
| Yelp Reviews | ✕ | ✕ | ✕ | ✕ | ✕ | 0 |
| Laptop Indian Prices | ✕ | ✕ | ✕ | ✕ | ✕ | 0 |
| Beer Ratings | ✕ | ✕ | ✕ | ✕ | ✕ | 0 |
| Coffee Review | ✕ | ✕ | ✕ | ✕ | ✕ | 0 |
| Ramen Ratings | ✕ | ✕ | ✕ | ✕ | ✕ | 0 |
| Airbnb Seattle | ✕ | ✕ | ✕ | ✕ | ✕ | 0 |
| Company Employee Size | ✕ | ✕ | ✕ | ✕ | ✕ | 0 |
| Anime Planet Rating | ✕ | ✕ | ✕ | ✕ | ✕ | 0 |
| FilmTV Movie Rating | ✕ | ✕ | ✕ | ✕ | ✕ | 0 |
| Movies Dataset Revenue | ✕ | ✕ | ✕ | ✕ | ✕ | 0 |
| NBA Draft VORP | ✕ | ✕ | ✕ | ✕ | ✕ | 0 |
| Mercedes Italy Cars | ✕ | ✕ | ✕ | ✕ | ✕ | 0 |

# F. Image-Tabular Curation

### F.1. Existing Benchmarks

The image-tabular benchmarking landscape is substantially more limited than its text-tabular counterpart. MuG (Lu et al., 2023) reports 8 text-image-tabular datasets, but these correspond to only 4 underlying datasets, some of them using different target variables. Tang et al. (2024b) curate 22 datasets spanning varying modality combinations: 6 are text-tabular and overlap with text-tabular benchmarks; 5 are text-image datasets that lie outside the scope of this paper; and the remaining 11 qualify for our image-tabular definition (6 of them also have text). In addition, we include the datasets introduced by TIME (Luo et al., 2025b) and MultimodalTabPFN (Kim et al., 2025b), some of them overlapping with aforementioned benchmarks.

However, many of these datasets suffer from serious reproducibility problems. For example, the *Seattle* dataset contains links to images via external URLs that are no longer reachable; and the *KARD* dataset points to a Kaggle dataset that has since been deleted. The remaining candidates are partially recoverable, but their preprocessing logic is often undocumented

and difficult to replicate faithfully.

After deduplication and removal of unavailable datasets, we are able to evaluate 16 unique datasets, from which only 5 pass the curation filter. For the ones which did non pass, it was sometimes hard to assess whether we have curated them properly. Therefore, we do not report curation statistics at the same level of detail as for text-tabular, and focus the remaining of the section on elaborating on the curation process.

### F.2. Curation Logic

The curation of datasets found in the wild involved several decisions with the goal of making the image feature important and interesting enough to qualify as a relevant true image-tabular task.

**Images.** Each dataset contains exactly one image column; datasets with multiple image fields per row (e.g., product galleries) were reduced to a single image for simplicity. Rows with absent or corrupt image files are dropped without imputation, as there is no sensible substitute for a missing image, and placeholder images would inject noise into the encoding step.

**Feature and Target engineering.** In several cases the raw target required transformation before satisfying the curation criteria, and we provide a non-exhaustive list of examples. *Log transformation*: Amazon Bestseller retail price is transformed as $\log(1 + \text{price})$ to stabilize the regression target across several orders of magnitude. *Quantile binning*: CS:GO Skin Price (10 equal-frequency bins), PetFinder listed age (8 bins), and HubMAP HPA donor age (10 bins) are discretized into multiclass targets. *Feature removal*: structured columns that directly encode the target or fully dominate the image signal are dropped. This is particularly evident in examples like Zooscan Plankton, where features were extracted directly from the image, and removing them increased the image importance.

**Kaggle upload.** To ensure reproducibility, all 20 image-tabular datasets are preprocessed and uploaded to Kaggle under the MulTaBench organization. Each upload contains a flat `images/` directory with one file per row named consistently, a `data.csv` with features and target. The image column stores relative paths into `images/` directory. A unified loading API handles download and ingestion, ensuring all datasets are accessed identically regardless of original source format.

## G. Text-Image-Tabular Datasets

From the 20 image-tabular datasets in MulTaBench, 8 of them include one or more text columns alongside the image and structured features. To investigate whether this could be treated as true text-image-tabular datasets, we apply also the full text curation pipeline to each, by conducting the independent test elaborated on Appendix C.3 to both image and text. By applying the selection rule independently, we prove that the 3 modalities contribute to the prediction to fulfill the *Joint Signal* criterion. In addition, for *Task-awareness*, we require that TAR on the image and on the text would improve on the respective frozen conditions. Finally, we also explicitly demand that performing TAR over both modalities (i.e., finetuning both the image and text encoder, separately) would improve on finetuning only one of them.

Of the 8 candidates, we find that only two satisfy all criteria for at least 3 learners: *PetFinder* and *Amazon Packages*. The remaining 6 fail primarily because text TAR does not improve over the frozen joint baseline, which might be a relative strict requirement. For future text-image-tabular efforts, one could consider relaxing this last condition, by only demanding that at least one of the modalities gains from representation tuning.

*Table 6.* The PetFinder Analysis. S=Structured, I=Image, T=Text. For all models, performing Joint Modeling and Target-Aware Representations for both modalities maximizes AUC (shown in %).

| Model | Single modality | | Frozen combinations | | | Target-Aware Representations (TAR) | | |
|---|---|---|---|---|---|---|---|---|
| | I | T | S+I | S+T | S+I+T | S+I$_{TAR}$+T | S+I+T$_{TAR}$ | S+I$_{TAR}$+T$_{TAR}$ |
| LightGBM | 77.2 | 72.1 | 79.9 | 77.7 | 81.1 | 82.8 | 84.2 | **85.7** |
| CatBoost | 78.9 | 73.5 | 81.7 | 79.3 | 83.2 | 83.9 | 85.2 | **86.4** |
| TabM | 80.2 | 74.9 | 83.0 | 80.7 | 84.2 | 84.8 | 86.3 | **87.0** |
| TabPFNv2 | 80.7 | 73.5 | 83.2 | 79.3 | 83.9 | 84.5 | 86.3 | **87.1** |
| TabPFN-2.5 | 81.1 | 76.0 | 83.7 | 81.0 | 84.9 | 85.3 | 87.3 | **88.0** |

Table 6 presents a case study of this analysis for the *PetFinder* dataset., and Table 7. does the same for the *Amazon Packages* dataset.

*Table 7.* Amazon Packages Analysis. S=Structured, I=Image, T=Text. Mean $R^2$ (%) per model and condition. For all models, TAR over both modalities dominates.

| Model | Single modality | | Frozen combinations | | | Target-Aware Representations (TAR) | | |
|---|---|---|---|---|---|---|---|---|
| | I | T | S+I | S+T | S+I+T | S+I$_{TAR}$+T | S+I+T$_{TAR}$ | S+I$_{TAR}$+T$_{TAR}$ |
| LightGBM | 43.2 | 17.5 | 45.4 | 20.8 | 49.9 | 56.9 | 52.2 | **59.4** |
| CatBoost | 44.2 | 19.0 | 46.7 | 22.5 | 52.5 | 58.1 | 53.8 | **59.8** |
| TabM | 46.6 | 20.5 | 48.6 | 24.2 | 55.7 | 60.3 | 56.1 | **61.1** |
| TabPFNv2 | 46.6 | 20.2 | 49.3 | 23.9 | 54.9 | 59.8 | 56.2 | **61.3** |
| TabPFN-2.5 | 47.5 | 21.1 | 50.2 | 24.9 | 56.2 | 60.7 | 57.2 | **61.5** |

# H. Extended Results

## H.1. Main Results Breakdown

**New Models.**   We extend our models suite by adding new models.

- For XGBoost, we follow previous work (Gorishniy et al., 2021; Arazi et al., 2025) and use the default implementation from the *xgboost* package,[7] with $booster = gbtree, early\_stopping\_rounds = 50, n\_estimators = 2000$.

- For RandomForest, we use the default scikit-learn implementation with default configuration with $n\_estimators = 100$.

- For *RealMLP* we use its official implementation in the *pytabkit* package, disable label smoothing and optimize for cross_entropy for binary classification and $1 - auc\_ovr$ for multiclass classification, keeping the other default hyperparameters.

- For *TabICLv2,*[8] *TabSTAR,*[9] *ConTextTab,*[10] and *TabDPT,*[11] we use their default implementations.

- For *AutoGluon-Multimodal* we use *MultiModalPredictor*[12] with $pretrained = True$ and optimizing for $roc\_auc$ (binary classification), $roc\_auc\_ovr$ (multiclass classification), and $r^2$ (regression).

**Task Type Breakdown**   Figures 7 and 8 replicate Figure 2, but breaking down to classification and regression datasets respectively. TAR consistently outperforms Frozen in both task types and both modalities, indicating that the benefit of target-aware representations is not specific to any of them.

**Win Rate by Model**   Table 8 reports the fraction of (dataset, fold) pairs where TAR outperforms Frozen for each model, with 95% CIs. End-to-end systems that do not expose a separate TAR condition for a given modality (TabSTAR, ConTextTab) are excluded from the corresponding column. TAR beats Frozen in the large majority of runs across all models and both modalities.

**Per-dataset Results.**   Tables 9 and 10 report per-dataset results for all 20 image-tabular and 20 text-tabular datasets, averaged over all learners that have both Frozen and TAR conditions and 5 random seeds, sorted by TAR gain. Negative $R^2$ scores are clipped before averaging.

---

[7]https://pypi.org/project/xgboost/
[8]https://pypi.org/project/tabicl/
[9]https://pypi.org/project/tabstar/
[10]https://github.com/SAP-samples/sap-rpt-1-oss
[11]https://pypi.org/project/tabdpt/
[12]https://auto.gluon.ai/stable/api/autogluon.multimodal.MultiModalPredictor.html

*Table 8.* Per-model TAR win rate on MulTaBench. End-to-end models excluded from columns where they lack a separate TAR condition.

| Model | Image (%) | Text (%) | All (%) |
|---|---|---|---|
| CatBoost | $90.0 \pm 5.9$ | $93.0 \pm 5.0$ | $91.5 \pm 5.5$ |
| LightGBM | $84.0 \pm 7.2$ | $93.0 \pm 5.0$ | $88.5 \pm 6.2$ |
| RF | $85.0 \pm 7.0$ | $90.0 \pm 5.9$ | $87.5 \pm 6.5$ |
| XGBoost | $80.0 \pm 7.8$ | $90.0 \pm 5.9$ | $85.0 \pm 6.9$ |
| TabPFNv2 | $77.0 \pm 8.2$ | $84.4 \pm 7.5$ | $80.7 \pm 7.9$ |
| TabM | $82.0 \pm 7.5$ | $77.0 \pm 8.2$ | $79.5 \pm 7.9$ |
| TabDPT | $70.0 \pm 9.0$ | $87.0 \pm 6.6$ | $78.5 \pm 7.9$ |
| RealMLP | $78.0 \pm 8.1$ | $78.0 \pm 8.1$ | $78.0 \pm 8.1$ |
| TabSTAR | $76.0 \pm 8.4$ | — | $76.0 \pm 8.4$ |
| TabPFN-2.5 | $77.0 \pm 8.2$ | $73.3 \pm 9.1$ | $75.2 \pm 8.7$ |
| ConTextTab | $73.0 \pm 8.7$ | — | $73.0 \pm 8.7$ |
| TabICLv2 | $55.0 \pm 9.8$ | $75.0 \pm 8.5$ | $65.0 \pm 9.2$ |

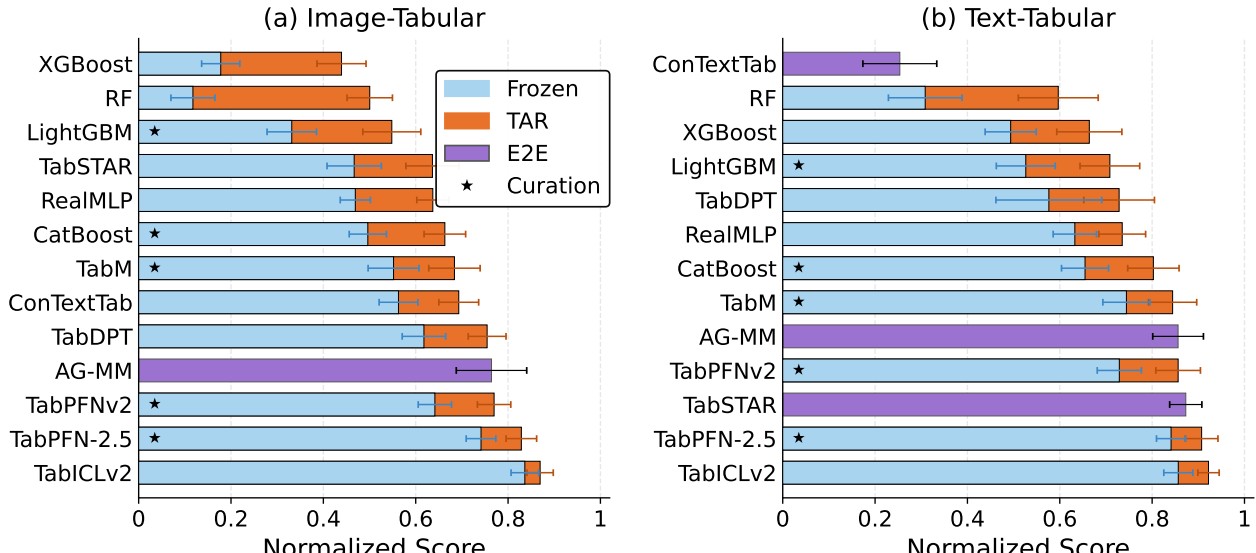

*Figure 7.* Tabular Learners Performances Analysis for Classification Tasks. Normalized scores over MulTaBench, with $\pm$ 95% CI.

## H.2. Missing Baselines

We deliberately exclude autoregressive generative models (LLMs and VLMs) from the benchmark evaluation, due to prohibitive inference costs and a memorization risk. The research on benchmarking LLMs and VLMs for MMTL task still needs to be explored. Although TIME (Luo et al., 2025b) and MultimodalTabPFN (Kim et al., 2025b) are relevant baselines, TIME has not released the code at the time of our submission. MultimodalTabPFN, on the contrary, has a working codebase, but it is highly not flexible to serve the model using the popular *scikit-learn* (Pedregosa et al., 2011) wrapper, making it hard to evaluate.

## H.3. Computation Costs

Table 11 and Figure 9 reports median wall-clock runtimes and peak GPU memory per (dataset, fold) run on a single *NVIDIA A100-SXM4* GPU with 40GB memory, and 8 CPU cores of type *AMD EPYC 7742 64-Core Processor*. We report results for each of the 5 core learners across frozen and TAR conditions and both encoder sizes. From the table, it is evident that the embeddings dominate all metrics. TAR adds a substantial overhead relative to frozen embeddings, dominated by the encoder fine-tuning step. For image datasets with the small DINO encoder, TAR roughly doubles runtime; the large encoder raises costs further.

Text TAR is significantly more expensive: *e5-small* TAR takes roughly ten times longer than frozen, and *e5-large* TAR approaches three hours per run. The gap arises partly as text-tabular datasets often contain more than a single text column,

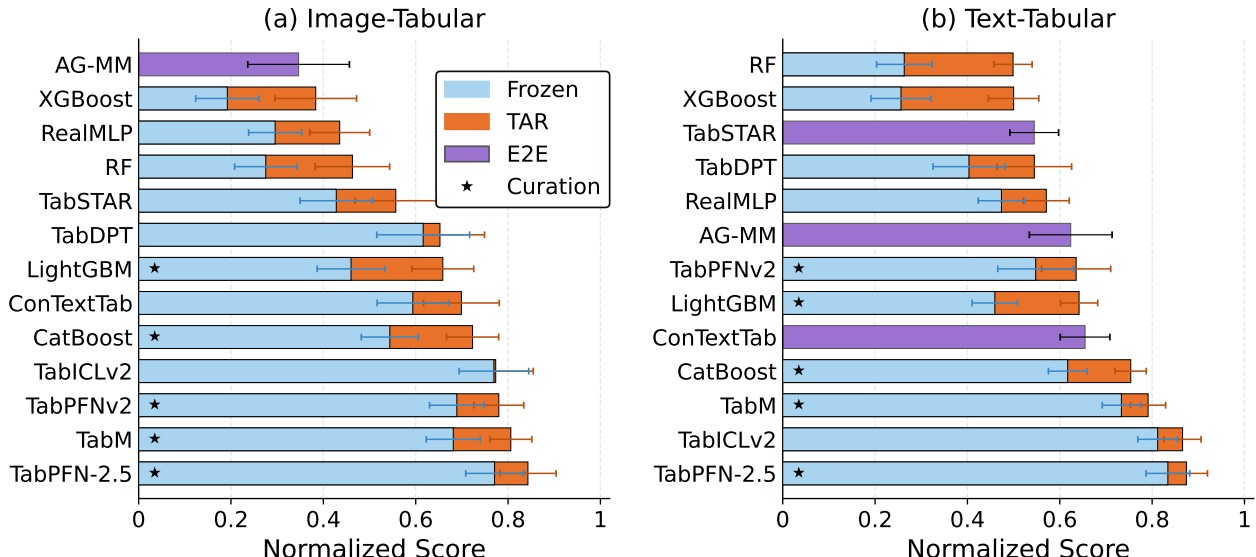

*Figure 8.* Tabular Learners Performances Analysis for Regression Tasks. Normalized scores over MulTaBench, with $\pm$ 95% CI.

making their effective dataset size much bigger.

The costs above are measured without any hyperparameter optimization (HPO). Standardizing HPO across 40 datasets is computationally prohibitive under the TAR paradigm: the encoder must be fine-tuned separately for each cross-validation fold to prevent data leakage, so a standard HPO sweep would require repeating encoder fine-tuning for every hyperparameter trial, multiplying an already expensive operation by the number of trials. Consequently, all experiments use a single fixed LoRA configuration across all datasets, with no per-dataset tuning of the encoder or the learner. All reported gains should therefore be interpreted as conservative lower bounds on what a fully tuned system could achieve.

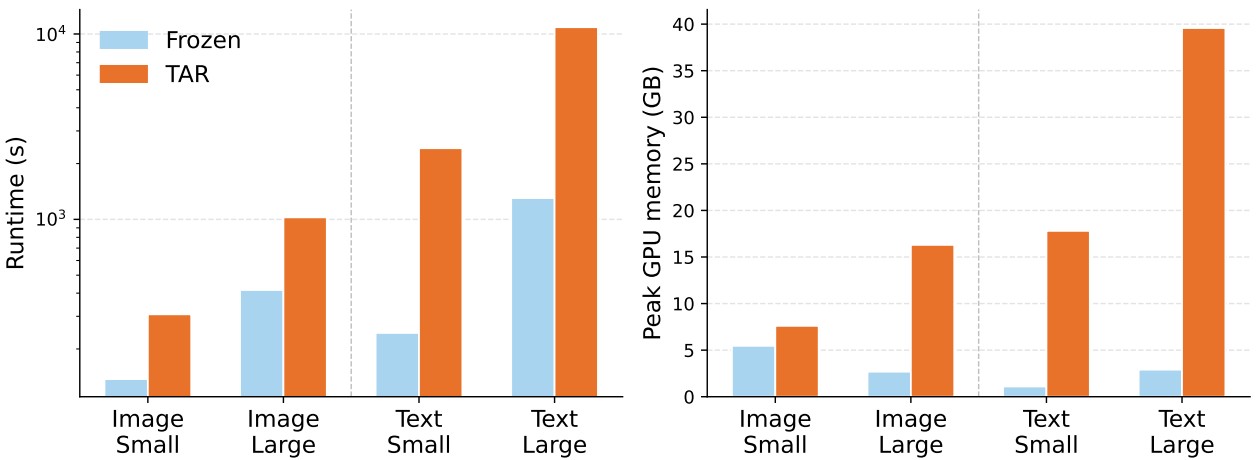

*Figure 9.* Computation costs per run. Left: median runtime in seconds (log scale). Right: median peak GPU memory. The dashed vertical line separates image (left) and text (right) conditions.

### H.4. Encoder Scale by Task Type

We replace our default encoders, DINO-small and E5-small, with DINO-large[13] and e5-large. Roughly speaking, this moves from models of 30M paremeters to models of 300M parameters. We then re-evaluate all 20 image and 20 text datasets.

---

[13]The official names are *dinov3-vits16-pretrain-lvd1689m* for small, and *dinov3-vitl16-pretrain-lvd1689m* for large.

*Table 9.* MulTaBench Image-Tabular Per-dataset Results. Averaged over 12 learners and 5 seeds, with both Frozen and TAR conditions, sorted by Gain. AUROC for classification, $R^2$ for regression.

| Dataset | Frozen | TAR | Gain |
|---|---|---|---|
| Mango Mass | 0.533 | 0.653 | +0.120 |
| Khaadi Clothes | 0.565 | 0.683 | +0.118 |
| Amazon Packages | 0.523 | 0.579 | +0.056 |
| CheXpert | 0.762 | 0.803 | +0.041 |
| CBIS-DDSM | 0.858 | 0.893 | +0.035 |
| Amazon Bestseller | 0.543 | 0.566 | +0.024 |
| MkPhoto Bots | 0.341 | 0.361 | +0.020 |
| Justin Instagram | 0.956 | 0.974 | +0.017 |
| Hateful Meme | 0.745 | 0.760 | +0.015 |
| H&M Fashion | 0.389 | 0.404 | +0.015 |
| Glaucoma SMDG | 0.921 | 0.934 | +0.012 |
| Mammography CMMD | 0.770 | 0.780 | +0.010 |
| CelebA Attractiveness | 0.903 | 0.913 | +0.009 |
| PetFinder | 0.832 | 0.841 | +0.009 |
| HubMAP HPA | 0.959 | 0.968 | +0.009 |
| Flower Bouquets | 0.627 | 0.636 | +0.009 |
| Painting Price | 0.270 | 0.275 | +0.005 |
| Letterboxd Movies | 0.439 | 0.443 | +0.004 |
| Zooscan Plankton | 0.983 | 0.985 | +0.003 |
| CS:GO Skins | 0.871 | 0.870 | −0.001 |
| *Mean* | 0.682 | 0.703 | +0.022 |

Figures 11 and 12 replicate Figure 10 restricted to classification and regression datasets respectively. TAR consistently outperforms Frozen across encoder sizes and task types, confirming that the benefit of TAR generalizes also to larger encoders.

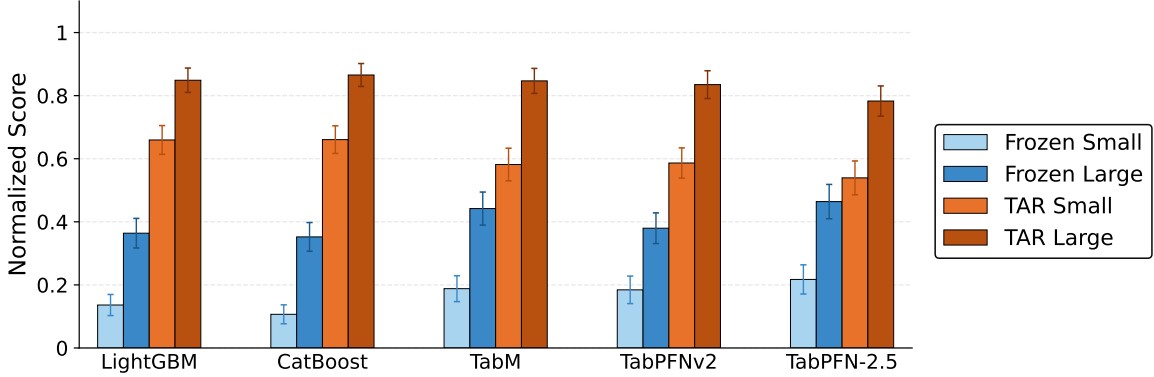

*Figure 10.* Embedding Model Size Analysis. Normalized scores are computed with min-max scaling at the learner level. TAR variants outperform the frozen ones for both model sizes.

## H.5. PCA Dimensions Analysis

To verify that the gains from target-aware adaptation do not depend on the PCA compression step, we repeat the core Frozen vs. TAR comparison using raw 384-dimensional embeddings, omitting the projection entirely. Since this largely increases the number of features for the downstream task, we limit the analysis to CatBoost and LightGBM, and exclude datasets with more than 5 text features, resulting in 33 datasets. Figure 14 shows 4 conditions side by side, varying between N=30 and No-PCA, and Frozen vs TAR. We observe that TAR outperforms Frozen in both settings, for both learners, confirming that the advantage is not an artifact of dimensionality reduction. The signal surfaced by fine-tuning is present in the raw 384-dimensional space and persists regardless of whether embeddings are subsequently compressed.

*Table 10.* MulTaBench Text-Tabular Per-dataset Results. Averaged over 10 learners and 5 seeds, with both Frozen and TAR conditions, sorted by Gain. AUROC for classification, $R^2$ for regression.

| Dataset | Frozen | Contextualized | Gain |
|---|---|---|---|
| Jigsaw Toxicity | 0.806 | 0.926 | +0.119 |
| Video Games Sales | 0.348 | 0.385 | +0.036 |
| Mercari | 0.412 | 0.436 | +0.024 |
| Baby Products | 0.873 | 0.895 | +0.022 |
| Vancouver Salaries | 0.753 | 0.775 | +0.022 |
| Rotten Tomatoes | 0.490 | 0.508 | +0.018 |
| Kickstarter | 0.720 | 0.737 | +0.017 |
| Book Price | 0.529 | 0.543 | +0.013 |
| Zomato Restaurants | 0.818 | 0.832 | +0.013 |
| Michelin Guide | 0.896 | 0.909 | +0.013 |
| Book Readability | 0.809 | 0.821 | +0.012 |
| Wine Review | 0.968 | 0.976 | +0.009 |
| Product Sentiment | 0.901 | 0.909 | +0.009 |
| SciMagojr Impact | 0.852 | 0.860 | +0.009 |
| US Accidents | 0.965 | 0.974 | +0.008 |
| Women's Clothing | 0.903 | 0.909 | +0.006 |
| Spotify Genres | 0.935 | 0.940 | +0.005 |
| Data Scientist Salary | 0.823 | 0.828 | +0.005 |
| Montgomery Salaries | 0.968 | 0.972 | +0.004 |
| Fake Job Postings | 0.916 | 0.918 | +0.002 |
| *Mean* | 0.774 | 0.792 | +0.018 |

## I. Additional Attention Maps

Each of the 4 datasets in Figure **??** is accompanied by 3 additional test-set examples below. In every case, Frozen attention remains scattered across task-irrelevant regions, while Target-Aware attention converges on semantically meaningful area identified in the main figure, relevant to the prediction.

## J. Discussion and Conclusion

In this work, we introduce MulTaBench, a benchmark of 40 image-tabular and text-tabular datasets designed to explore challenging Multimodal Tabular Learning tasks. We contribute the largest image-tabular benchmark to date, while focusing on tasks that benefit from Joint Modeling and TAR, differing ourselves from existing MMTL benchmarks. Our findings show that existing models rely on representations that are often insufficient for the task at hand, making MulTaBench a necessary tool for evaluating the next generation of Multimodal Tabular Foundation Models.

MulTaBench suffers from an important limitation: our curation pipeline entangles the computational problem with the algorithmic solution. As such, it is hard to predict in advance whether a new dataset meets our criteria, and the models used for the curation cannot be fairly evaluated due to selection bias. While we believe future research should aim to address these limitations, our work is a strong step to tackle a problem which was overlooked so far, yielding findings that generalize well to new models. Importantly, the automated nature of our pipeline facilitates the continuous expansion of MulTaBench to include new dataset candidates, the latest tabular learners, or refined selection logic as the field matures. As such, our curation pipeline is a contribution of its own, providing a mechanism to refresh the benchmark with harder candidates as current tasks become saturated by future models.

Our research paves the way to many exciting future directions, such as expanding to a dedicated text-image-tabular benchmark, exploring other modalities such as audio and videos, or analyzing different prompting strategies to steer embeddings towards the target. Mainly, MulTaBench supports the development of Multimodal TFMs. In our opinion, there are two big challenges to solve: architecture and training data. For architectures, in §5, we suggest that future models should ideally take the best out of ICL and finetuning; for instance, coupling TFMs with LLMs and VLMs is a compelling path. For training data, since real data corpora for MMTL are rare, (Eggert et al., 2023), expanding the syntethic numerical priors used for training TFMs (Hollmann et al., 2025; Zhang et al., 2025a; Qu et al., 2026) to include text and image features is an exciting direction (Luo et al., 2025a; Brahmavar et al., 2026). We hope that our work will contribute to the research of Multimodal Tabular Learning, and we are excited towards a future where this crucial problem sees the progress it deserves.

*Table 11.* Computation costs runs. Median Runtime in seconds and Median Peak GPU memory in GB. Partition by tabular learners, modality and encoder size.

| | Small Encoder | | | | Large Encoder | | | |
| | Runtime (s) | | Peak GPU (GB) | | Runtime (s) | | Peak GPU (GB) | |
| **Model** | Frozen | TAR | Frozen | TAR | Frozen | TAR | Frozen | TAR |
|---|---|---|---|---|---|---|---|---|
| *Image-Tabular (DINO encoder)* | | | | | | | | |
| LightGBM | 141 | 287 | 5.5 | 7.6 | 411 | 1,003 | 2.7 | 16.7 |
| CatBoost | 128 | 307 | 5.4 | 7.6 | 432 | 1,002 | 2.7 | 16.7 |
| TabM | 129 | 282 | 5.4 | 7.6 | 375 | 993 | 2.7 | 16.8 |
| TabPFNv2 | 173 | 318 | 5.3 | 7.6 | 432 | 1,055 | 2.7 | 14.8 |
| TabPFN-2.5 | 169 | 316 | 5.5 | 7.6 | 405 | 1,048 | 2.7 | 14.8 |
| *Text-Tabular (E5 encoder)* | | | | | | | | |
| LightGBM | 223 | 2,417 | 1.1 | 17.3 | 1,284 | 10,867 | 2.9 | 39.6 |
| CatBoost | 323 | 2,471 | 1.1 | 17.3 | 1,377 | 10,860 | 2.9 | 39.6 |
| TabM | 225 | 2,408 | 1.2 | 17.3 | 1,256 | 10,833 | 2.9 | 39.6 |
| TabPFNv2 | 253 | 2,430 | 1.1 | 18.6 | 1,368 | 11,569 | 2.9 | 39.6 |
| TabPFN-2.5 | 242 | 2,396 | 1.1 | 18.6 | 1,347 | 11,556 | 2.9 | 39.6 |

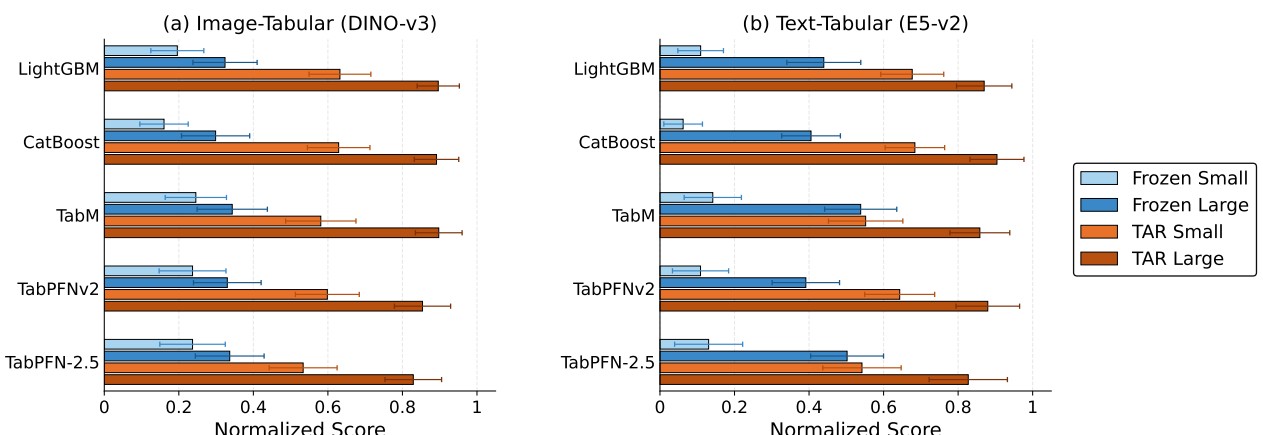

*Figure 11.* Encoder Scale Analysis for Classification. Small and large encoder variants, frozen and TAR, normalized within each model.

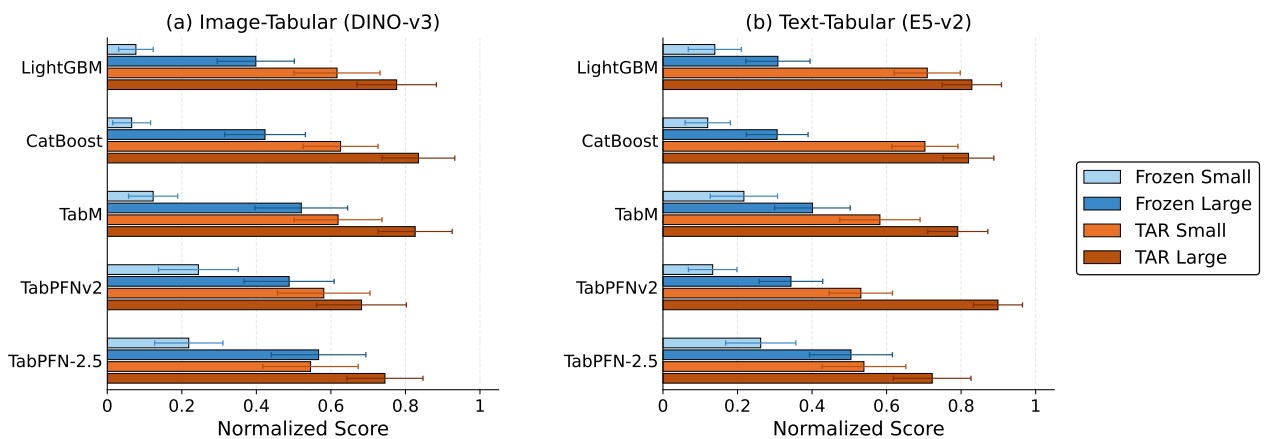

*Figure 12.* Encoder Scale Analysis for Regression. Small and large encoder variants, frozen and TAR, normalized within each model.

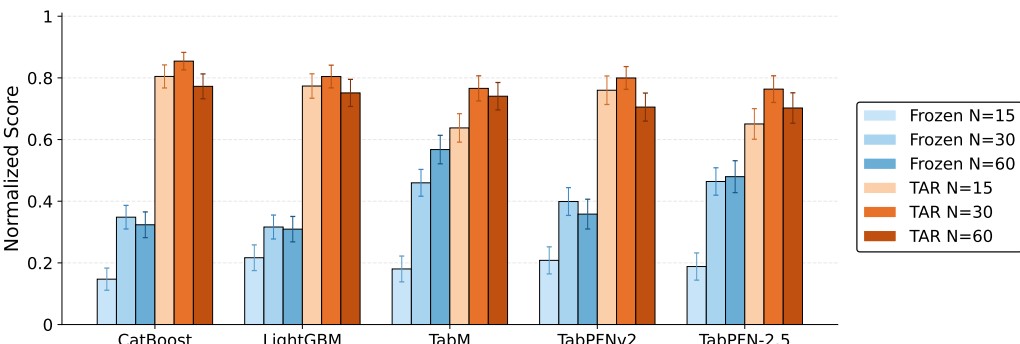

*Figure 13.* Embedding Dimension Analysis. Normalized scores are computed with min-max scaling at the learner level. TAR variants are stronger than Frozen ones for 15, 30, and 60 PCA components.

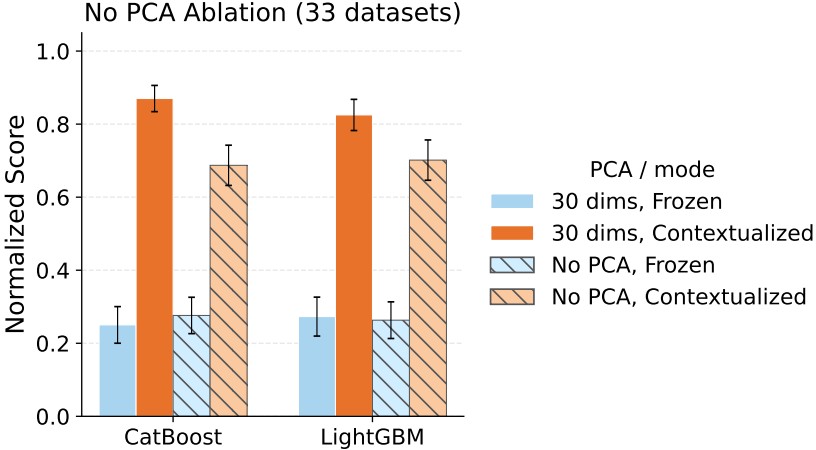

*Figure 14.* No-PCA ablation on 33 datasets for CatBoost and LightGBM. Normalized scores are on the model level.

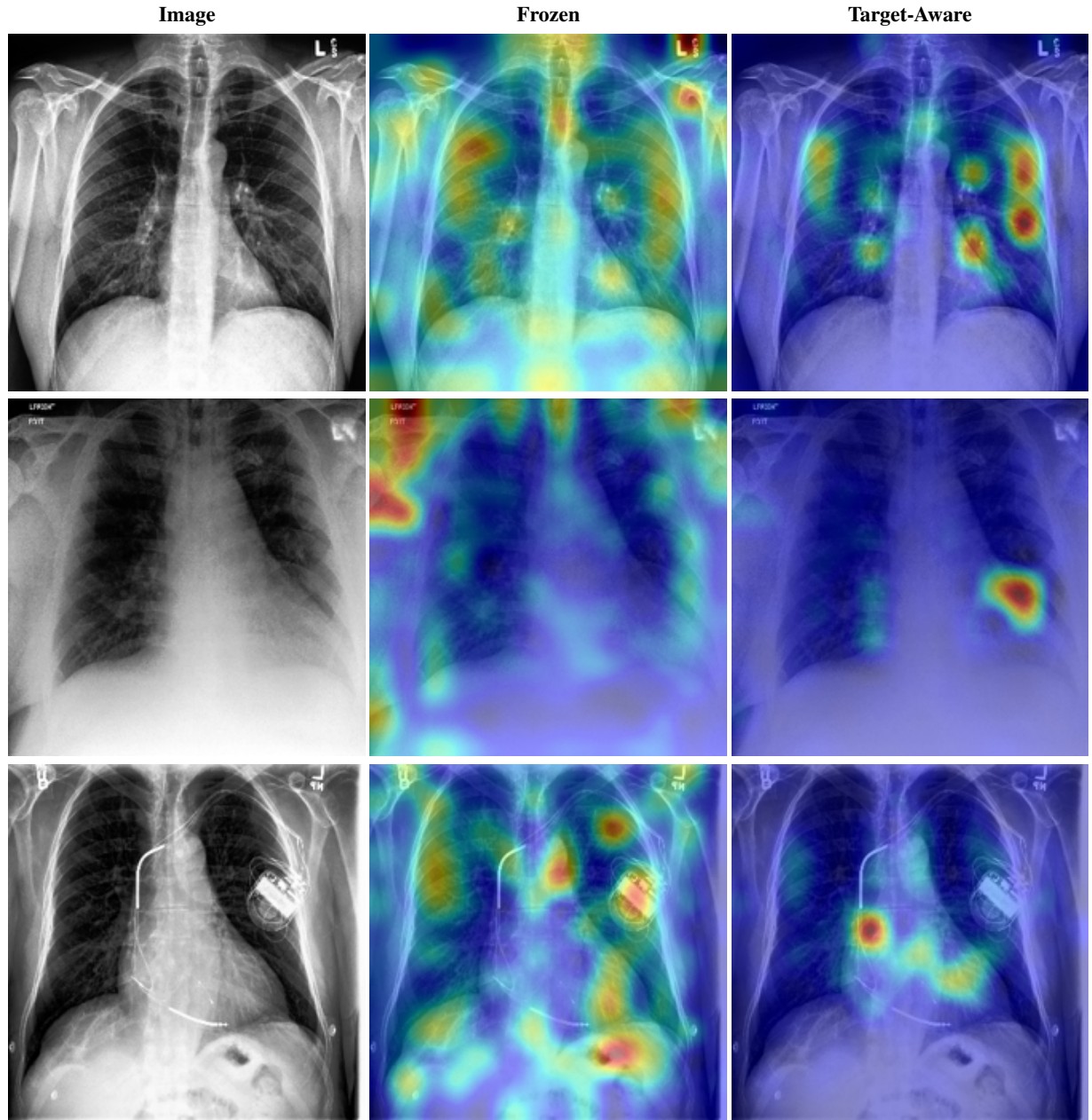

*Figure 15.* CheXpert Attention Maps. The attention shifts from diffused edges to the lung.

| Image | Frozen | Target-Aware |
|:-:|:-:|:-:|

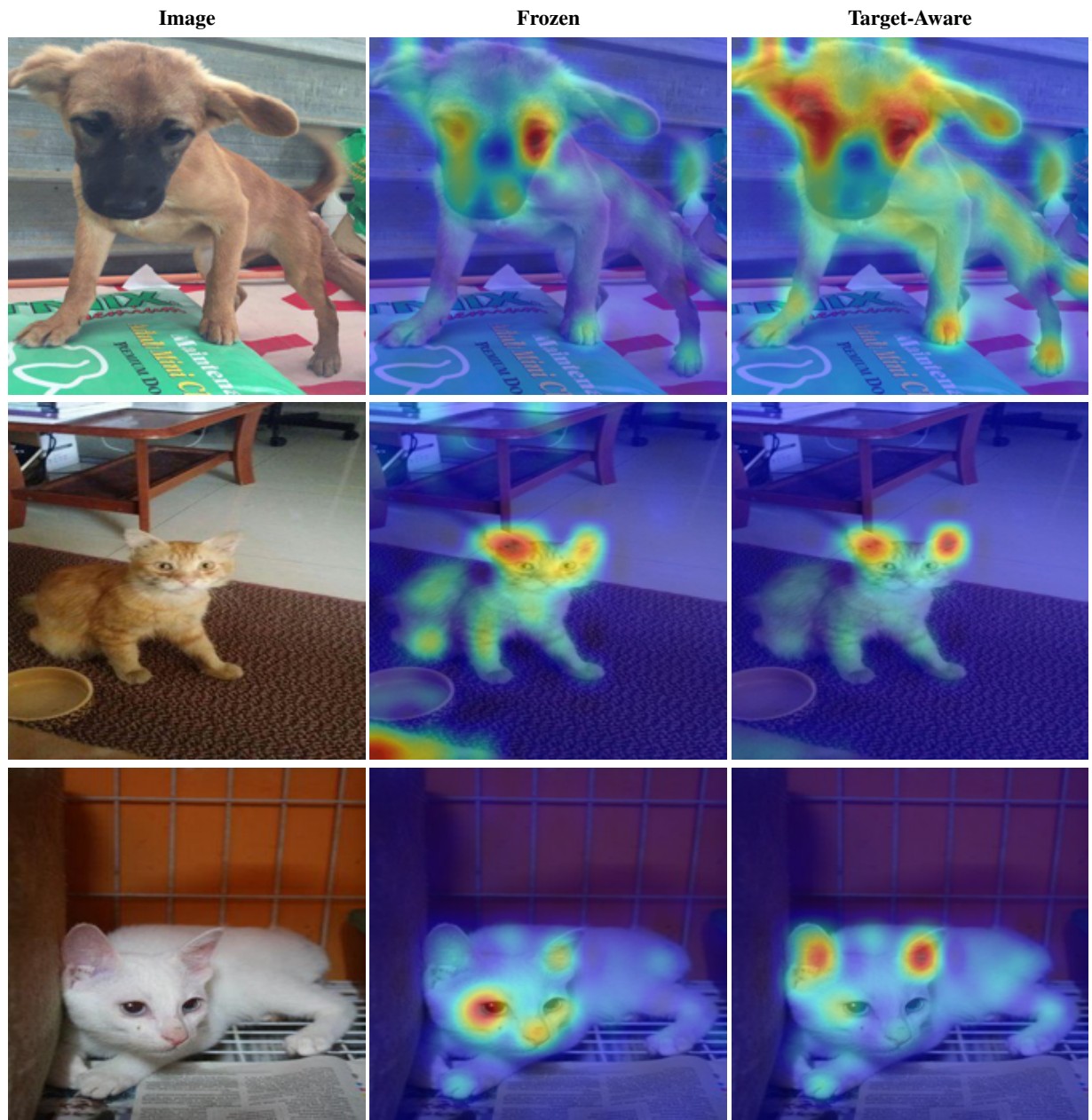

*Figure 16.* PetFinder Attention Maps. Attention isolates the cat ears and the dog's eyes.

| Image | Frozen | Target-Aware |
|:-----:|:------:|:------------:|

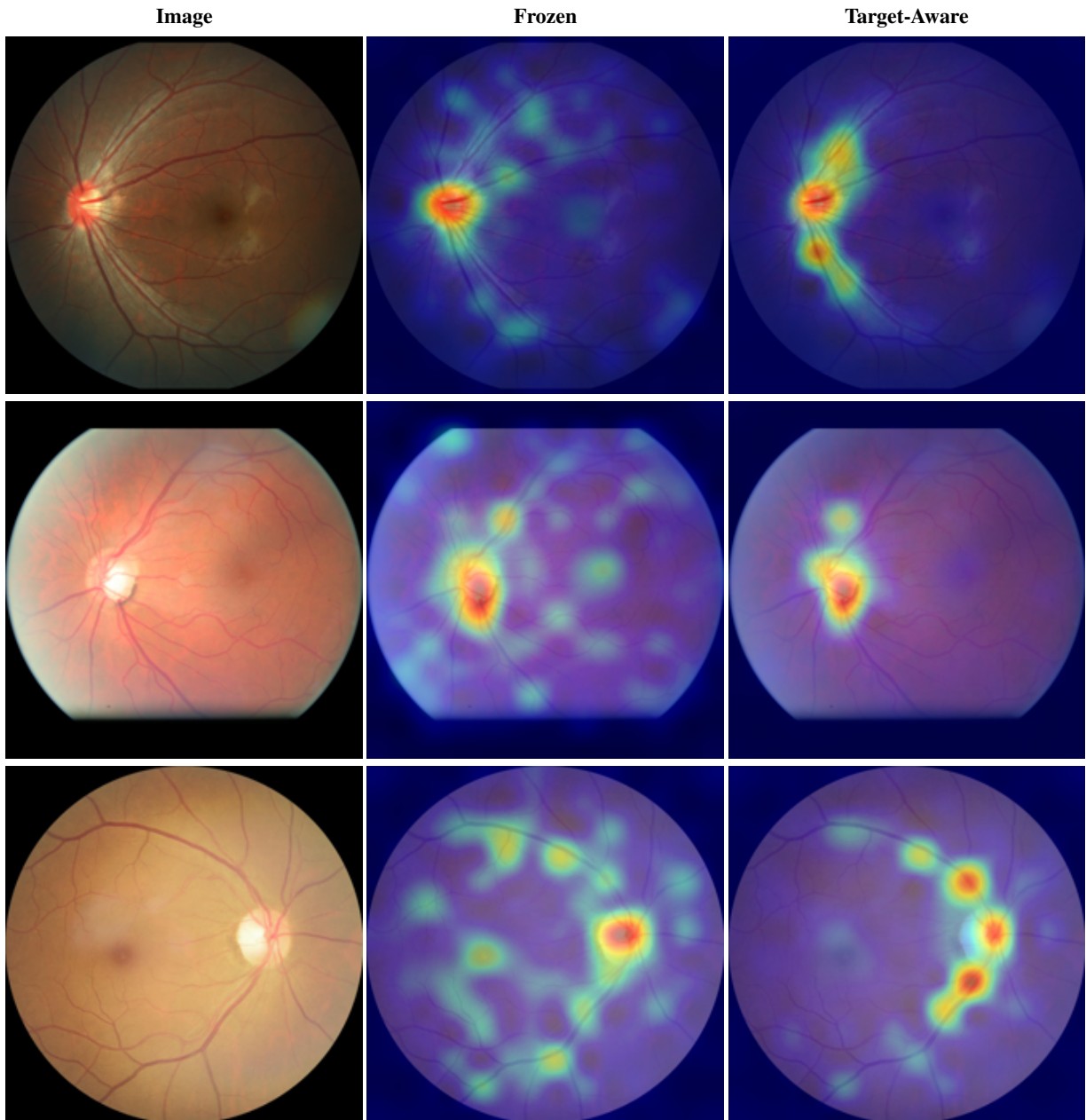

*Figure 17.* Glaucoma Attention Maps. Frozen attention scatters randomly across the retina; TAR converges on the optic disc and nerve fiber region, the clinically relevant area for glaucoma diagnosis.

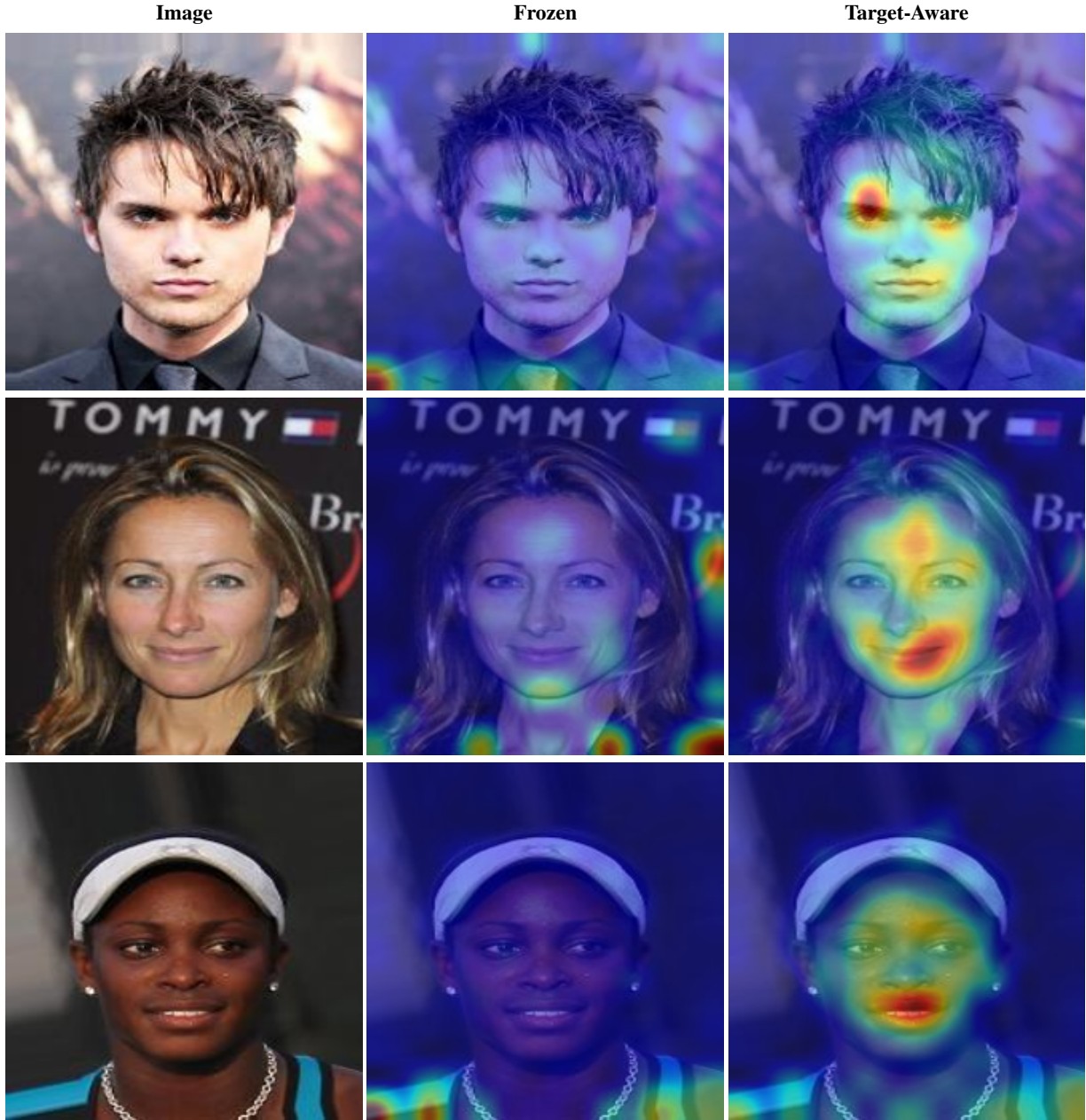

*Figure 18.* Celeb Attractiveness Attention Maps. Frozen attention disperses across accessories, clothing, and background; TAR consistently focuses on facial features.

