# OpenReview forum: "MulTaBench: Benchmarking Multimodal Tabular Learning with Text and Image"
_ICML.cc/2026/Workshop/FMSD — FMSD @ ICML 2026 Poster_

### Official Review · Reviewer_tLjo · 2026-05-15
**A Relevant Benchmark for Multimodal Tabular Foundation Models**

**Rating:** 7
**Confidence:** 4

**Review:**

This paper introduces MulTaBench, a benchmark for multimodal tabular learning with text and image modalities. The main motivation is that current tabular foundation models are strong on structured numerical data, but they do not natively support unstructured modalities such as text and images. The paper argues that simply using frozen pretrained embeddings is often insufficient, because generic embeddings may discard fine-grained, task-relevant information. To address this, the authors introduce the notion of Target-Aware Representations (TAR), where text or image encoders are adapted to the prediction target before being used by tabular learners.


Strengths:

The paper is highly relevant to the workshop. Multimodal structured data is an important direction for tabular foundation models, and the paper clearly identifies a limitation of current TFMs: they are mostly designed for numerical or structured inputs and rely on external frozen embeddings for text and images. This is a meaningful gap, especially for real-world applications where tables often come with product descriptions, images, medical scans, social media content, or other unstructured fields.

The benchmark contribution is useful. The paper does not simply collect datasets with multiple modalities, but tries to curate tasks where both modalities matter and where frozen embeddings are not enough. This makes the benchmark more targeted than a generic multimodal dataset collection. I also appreciate the balanced design of the benchmark, with 20 image-tabular and 20 text-tabular datasets, as well as both classification and regression tasks.

The authors evaluate several tabular learners, including both classical models and tabular foundation models, and show that target-aware representations consistently improve over frozen embeddings. The robustness checks over encoder scale and embedding dimension are also helpful, since they reduce the concern that the gains are merely due to using small encoders or aggressive PCA compression.

The discussion of future multimodal TFMs is also valuable. The paper makes a good case that the next generation of tabular foundation models should combine the robustness and efficiency of in-context tabular learners with the ability to adapt unstructured representations to the target task.

Areas for Improvement:

The main weakness is that the curation criterion is somewhat tied to the proposed solution. A dataset is included partly because TAR improves over frozen embeddings. This makes sense for constructing a benchmark focused on target-aware representations, but it also introduces a selection bias: the benchmark is naturally favorable to methods that perform this kind of adaptation. The authors acknowledge this limitation, but it should be emphasized more clearly in the main paper.

Another concern is that TAR is treated as a preprocessing step, but it still involves supervised finetuning on the target. This raises practical and methodological questions. For example, the computational cost can be high, especially for text datasets with multiple text fields, and the comparison with frozen embeddings may depend on the specific LoRA setup, training budget, and validation procedure. It would be useful to discuss more explicitly how this differs from training a full multimodal model end-to-end.

The benchmark would also benefit from clearer information about dataset availability, licenses, and reproducibility. The image-tabular datasets are especially valuable, but some of them appear to involve substantial preprocessing and manual curation. For a benchmark paper, the long-term usefulness depends heavily on whether the datasets, preprocessing scripts, and evaluation code are easy to access and reproduce.

Detailed Comments:

1. The distinction between Joint Signal and Task-awareness is useful, but the empirical acceptance rule could be better justified. In particular, the threshold δ = 0.001 and the requirement that 3 out of 5 learners pass seem somewhat arbitrary. A sensitivity analysis with stricter thresholds would help show that the selected datasets are stable.

2. Since the benchmark is curated based on TAR gains, models using TAR have an inherent advantage. The paper should be careful not to present the benchmark as a neutral ranking benchmark for all multimodal tabular learners. It is better framed as a benchmark for studying tasks where target-aware unstructured representations are needed.

3. The computational cost of TAR is non-trivial. The appendix reports that text TAR can be much more expensive than frozen embeddings, especially with large encoders. This issue deserves more attention in the main paper, because it affects whether TAR is practical for ordinary tabular learning workflows.

4. The paper evaluates many tabular learners, but some relevant multimodal baselines are missing, including LLM/VLM-style approaches and recent multimodal PFN-style models. I understand the cost and engineering constraints, but the absence of these baselines should be discussed more directly.

5. The qualitative attention maps are interesting and help illustrate the intuition behind target-aware adaptation. However, they are only suggestive. The paper should avoid over-interpreting them as strong evidence that TAR always focuses on semantically correct regions.

6. For text-tabular datasets, it would be useful to distinguish between different types of text fields, such as short categorical strings, IDs, product names, long descriptions, and user reviews. These likely require different modeling strategies.

7. The paper could make the future benchmark protocol clearer. For example, will future methods be allowed to finetune encoders on the training labels? Should all models use the same train/validation splits and the same target-aware preprocessing budget? These details matter for fair comparison.

---

### Official Review · Reviewer_krLy · 2026-05-16
**Introduction of an interesting new benchmark for multimodal tabular tasks**

**Rating:** 7
**Confidence:** 4

**Review:**

Summary
The authors introduced a new multimodal tabular benchmark with a creation of a curated dataset focus on cases where multimodal input is required. Through multiple experiments, they prove that having dedicated finetuned backbones for other modalities outperforms the frozen model counter part

Strengths
Introduction of a new curation pipeline that include datasets with joint signal and sask-awareness
Evaluation of multiple models to prove the need of task aware finetuning of text and image modalities
Good qualitative results through attention visualizations

Areas or improvement

The benchmark defines “good” datasets as those where LoRA-based target-aware tuning improves over frozen embeddings. This creates a circular dependency:
the benchmark favors datasets where the proposed adaptation method works
datasets where frozen embeddings are already sufficient are discarded
datasets where another adaptation strategy works better may also be excluded


Justification of the score:

The paper addresses an important and emerging problem in multimodal tabular learning and introduces a benchmark that is both timely and practically useful. The core idea, namely that many multimodal tabular datasets do not actually require target-aware multimodal reasoning, is interesting and well motivated. The paper is also generally clear and supported by extensive experimentation across multiple learners, modalities, embedding scales, and adaptation settings.

A major strength is the proposed curation pipeline based on Joint Signal and Task-awareness, which provides a more principled way to filter multimodal datasets compared to existing benchmarks. The benchmark itself is substantial in scale and diversity, especially on the image-tabular side, where benchmarking efforts are still limited. The empirical results consistently show that target-aware finetuning improves over frozen representations across many settings, which makes the findings convincing and likely useful for future multimodal TFM research.

At the same time, I believe the paper has several conceptual limitations that prevent it from being a top-tier benchmark paper. The main concern is that the benchmark definition is tightly coupled to the proposed LoRA-based adaptation setup. Since datasets are selected based on whether the proposed adaptation improves performance, the benchmark may inherently favor the authors’ hypothesis and exclude tasks where other forms of multimodal reasoning are relevant. As a result, the notion of “task-awareness” is not fully disentangled from the specific adaptation strategy used during curation.

Despite these concerns, I believe the paper makes a meaningful contribution. The benchmark is likely to be useful to the community, the experiments are extensive, and the paper raises important questions about the limitations of frozen multimodal representations for tabular learning. Overall, I believe the strengths outweigh the weaknesses, which justifies a score of 7 (Good paper, accept).

---

### Official Review · Reviewer_e9iP · 2026-05-21

**Rating:** 7
**Confidence:** 3

**Review:**

### Summary

The paper addresses limitations in current evaluation and modeling practices for multimodal tabular learning. Existing Tabular Foundation Models mainly handle structured data, while text and image inputs are typically represented using frozen pretrained embeddings.  The paper introduces MulTaBench, a benchmark of 40 datasets curated according to two criteria: Joint Signal, requiring the combination of structured and unstructured modalities to outperform either modality alone, and Task-awareness, requiring target-aware representations optimized for the prediction target to outperform frozen embeddings. Experiments show that target-aware representations consistently outperform frozen embeddings across different tabular learners, suggesting the importance of task-specific representation tuning for multimodal tabular prediction. However, since dataset selection partly depends on whether such tuning improves performance, the benchmark is somewhat coupled with the proposed evaluation method, making it difficult to determine in advance whether a new dataset satisfies the criteria.

### Strengths And Weaknesses

#### Strengths

- The paper accurately identifies a key research gap in current Tabular Foundation Models. Existing tabular foundation models perform strongly on structured tabular learning tasks, but they remain unimodal models and can only handle text and image inputs by relying on external frozen encoders.

- The paper does not simply treat every dataset containing both tabular and unstructured data as a valid multimodal tabular learning task. Instead, it explicitly proposes two core criteria: Joint Signal and Task-awareness. This distinction is highly practical, as it clearly separates multimodal prediction tasks from pure tabular tasks and NLP/CV tasks.

- The proposed dataset curation pipeline is clear, concise, and easy to understand. It translates the theoretical criteria into a concrete evaluation protocol with four experimental conditions. Compared with directly aggregating existing multimodal datasets, this benchmark construction process is more rigorous and scientifically grounded.

- The experimental results show that Target-Aware Representations consistently outperform frozen pretrained embeddings. This advantage appears not only in the models used for dataset curation, but also in other tabular learners and end-to-end multimodal models. Therefore, the conclusion is more convincing than one based only on a single learner or a single modality.

#### Weaknesses

- The dataset selection criterion partly relies on whether Target-Aware Representations improve performance over frozen embeddings. This means that MulTaBench is inherently biased toward task settings where the proposed target-aware representation preprocessing is effective. However, a benchmark should ideally define task difficulty independently of the specific algorithm used to verify that difficulty.

- Target-Aware Representations use target labels to tune the representations before training the tabular model, whereas the frozen embedding setting does not include this additional supervised adaptation stage. Therefore, the observed performance improvement may simply come from adding a supervised representation learning step, rather than purely demonstrating that frozen embeddings are intrinsically deficient. As a result, the empirical comparison does not constitute a fully fair controlled comparison between two equally trained pipelines.

- Although the encoder is tuned toward the prediction target, this adaptation process does not incorporate the structured tabular features. This means that the learned representations are aligned with the label information, but are not conditionally constrained by the tabular modality. Compared with the broader goal of multimodal tabular learning, where representations should ideally be aligned with both the prediction target and the other modalities, this design remains somewhat limited.